# Sea salt aerosol production via sublimating wind-blown saline snow particles over sea-ice: parameterizations and relevant micro-physical mechanisms

Xin Yang[1], Markus M. Frey[1], Rachael H. Rhodes[2*], Sarah J. Norris[3], Ian M. Brooks[3], Philip S. Anderson[4], Kouichi Nishimura[5], Anna E. Jones[1], Eric W. Wolff[2]

[1] British Antarctic Survey, Natural Environment Research Council, Cambridge, UK
[2] Department of Earth Sciences, University of Cambridge, Cambridge, UK
[3] School of Earth and Environment, University of Leeds, Leeds, UK
[4] Scottish Association for Marine Science, Oban, Argyll, Scotland, UK
[5] Graduate School of Environmental Studies, Nagoya University, Nagoya, Japan
*now at Department of Geography and Environmental Sciences, Northumbria University, Newcastle upon Tyne, UK

*Correspondence to*: Xin Yang (xinyang55@bas.ac.uk)

**Abstract.** Blowing snow over sea-ice has been proposed as a significant source of sea salt aerosol (SSA) (Yang et al., 2008). In this study, using snow salinity data and blowing snow and aerosol particle measurements collected in the Weddell Sea sea-ice zone (SIZ) during a winter cruise, we perform a comprehensive model-data comparison with the aim of validating proposed parameterizations. Additionally, we investigate possible physical mechanisms involved in SSA production from blowing snow. A global chemical transport model, p-TOMCAT, is used to examine the model sensitivity to key parameters involved, namely blowing snow size distribution, snow salinity, sublimation function, surface wind speed, relative humidity, air temperature and ratio of SSA formed per snow particle. As proposed in Yang et al.'s parameterizations, SSA mass flux is proportional to bulk sublimation flux of blowing snow and snow salinity. To convert bulk sublimation flux to SSA size distribution, requires (1) sublimation function for snow particles, (2) blowing snow size distribution, (3) snow salinity, and (4) ratio of SSA formed per snow particle.

The optimum model-cruise aerosol data agreement (in diameter range of 0.4-12 µm) indicates two possible micro-physical processes that could be associated with SSA production from blowing snow. The first one assumes that one SSA is formed per snow particle after sublimation, and snow particle sublimation is controlled by the curvature effect or the so-called 'air ventilation' effect. The second mechanism allows multiple SSAs to form per snow particle and assumes snow particle sublimation is controlled by the moisture gradient between the surface of the particle and the ambient air (moisture diffusion effect). With this latter mechanism the model reproduces the observations assuming that one snow particle produces ~10 SSA during the sublimation process. Although both mechanisms generate very consistent results with respect to observed aerosol number densities, they correspond to completely different micro-physical processes and show quite different SSA size spectra, mainly in ultra-fine and coarse size modes. However, due to the lack of relevant data, we could not, so far, conclude confidently which one is more realistic, highlighting the necessity of further investigation.

## 1 Introduction

Over most of the Earth, primary sea salt aerosol (SSA) derives from wave breaking and bubble bursting at the open ocean surface (e.g. de Leeuw et al., 2011). SSA is relevant to radiative forcing of climate because it can efficiently scatter solar radiation (O'Dowd et al., 1997; Murphy et al. 1998; Quinn et al., 2002). Moreover, SSA can serve as cloud condensation nuclei (CCN) (e.g. O'Dowd and Smith, 1993; O'Dowd et al., 1997; 1999) and even ice nucleating particles (INP) (Wise et al., 2012; DeMore et al., 2016) that influence global climate.

Observations of sulphate depletion relative to sodium in Antarctic aerosol and snow samples first argued for a sea-ice source of SSA (Wagenbach et al., 1998; Rankin and Wolff, 2003; Jourdain et al., 2008; Legrand et al., 2017). The depletion of sulphate is due to the effect of mirabilite ($Na_2SO_4.10H_2O$) precipitation from brine on sea-ice when temperature drops below -6.4°C (Bultler et al., 2016), a fractionation not plausible for sea spray particles generated directly from open ocean. Thus, it allows a new interpretation to the sodium recorded in ice cores, as open ocean sea spray is no longer the sole source for salts in snow and ice cores (e.g. Rankin et al., 2002; 2004). Moreover, this finding raises the possibility of using sea salt or sodium recorded in ice cores as a potential sea-ice extent proxy for past climates (Wolff et al., 2003; Abram et al. 2013; Severi et al., 2017).

Saline crystals on sea-ice, such as frost flowers (FFs) (e.g. Rankin et al. 2000, 2002; Kaleschke et al., 2004, Xu et al., 2016) with relatively high salinity and blowing snow (Yang et al., 2008) with relatively low salinity, were both suggested as potential sources of SSA. Evidence from laboratory chambers (Roscoe et al., 2011; Yang et al., 2017) and field measurements (Obbard et al., 2009; Hara et al., 2017) indicate that FFs are unlikely to be a major direct source. Global models with blowing snow as a SSA source implemented can successfully reproduce winter SSA peaks at high latitudes (Levine et al., 2014; Huang and Jaeglé, 2017; Rhodes et al., 2017). In addition, chemistry transport model studies demonstrate that when this sea-ice sourced SSA is treated as a source of bromine to the boundary layer, the polar springtime 'bromine explosion' events as well as the associated 'ozone depletion' events can be largely reproduced (Yang et al., 2010; Theys et al., 2011; Legrand et al. 2016; Zhao et al., 2016; 2017; Choi et al., 2018). However, the SSA production parameterisations implemented in models have not been fully validated against field data, and the possible physical mechanisms involved in the SSA formation are not completely clear.

In this study, based on a comprehensive set of measurements for both blowing snow particles and aerosol particles (Frey et al., 2019), made during a winter cruise on board the icebreaker *RV Polarstern* within the Weddell Sea ice zone (SIZ) in June-August, 2013, we could, for the first time, test and validate model parameterizations of SSA production, and investigate the model sensitivity to relevant parameters. A brief description of the cruise measurements is given in section 2. Parameterization and model experiments are detailed in section 3. Results of the model-data comparison are given in section 4. Relevant mechanisms of the SSA production from blowing snow are discussed in Section 5.

## 2 Measurements

The measurements used for the model validation were carried out during a winter sea ice cruise in the Weddell Sea, Antarctica aboard the German ice breaker *RV Polarstern* between 8 June and 12 August 2013 (Frey et al., 2019; Nerentorp Mastromonaco et al., 2016). The ship entered the sea ice zone on 17 June and penetrated into the Weddell Sea. From 20 July, the ship headed
back to the marginal sea ice zone before re-entering the pack ice again around the 24 July and finally returning to the open ocean on 9 August. The cruise track was such that a large part of the measurements was carried out during polar night, since the sun remained below the horizon between 23 June and 7 July providing only a few hours of twilight per day. A detailed description of instrumentation and measurement methods is given in an accompanying paper (Frey et al., 2019). In brief, airborne aerosol and suspended snow particle number concentrations at ambient temperature and humidity were continuously
measured from the crow's nest of the ship at 29 m above the sea surface at one-minute temporal resolution. Meteorological parameters were measured by the ship's meteorology observatory, and included air temperature and relative humidity at 29 m and wind speed and direction at 39 m above the sea ice surface (Figure 1b-d). Aerosol particles were detected using a Compact Light weight Aerosol Spectrometer (CLASP) and binned into 16 size bins covering the median diameter range from 0.36 to 11.62 μm (Hill et al., 2008; Norris et al., 2008). Suspended blowing snow particles were measured using a Snow Particle
Counter (SPC) described previously (Nishimura and Nemoto, 2005; Nishimura et al., 2014) with 64 size bins covering median diameters from 36 to 490 μm. Due to measurement uncertainty the SPC counts from the top and bottom bin are not used.

## 3 Model and parameterization of SSA from SIZ

### 3.1 Model set-up

Our global chemistry transport model, p-TOMCAT, has a detailed process-based SSA scheme (Levine et al., 2014). The
following updates have been introduced to this model in recent studies: more realistic model precipitation fields (Legrand et al. 2016); a sea spray emission scheme following the work of Jaeglé et al. (2011), and a modified surface snow salinity distribution function (Rhodes et al., 2017). Both open ocean sourced and sea-ice sourced SSA (as dry NaCl) are tagged in 21 size bins in size ranging 0.02-12 μm in order to track their relative contributions. For those ultra-fine particles (e.g. <0.1 μm), the below-cloud scavenging coefficient rates are taken from the Dick et al. (1990) scheme.
The meteorological forcing files for the model are 6 hourly reanalysis ERA-interim data from the European Centre for Medium-Range Weather Forecasts (ECMWF). Monthly sea-ice coverage and sea surface temperatures (SST) are taken from the Hadley Centre Sea Ice and Sea Surface Temperature (HadISST) dataset (Rayner et al., 2003). The model's horizontal resolution is 2.8° X 2.8° with 31 vertical layers from the surface to ~10 hPa at the top model layer. The bottom model layer has an average height of ~60 m. The spin-up time is >1 year to allow an equilibrium situation to be established. A three-year
integration (2013-2015) is used to obtain multi-year means.

The experiments carried out are summarised in Table 1. In the control run for sea-ice sourced SSA (SI_Base_A) a constant water mass loss rate against time for snow particle sublimation rate is assumed (see section 3.3.1) and mode A (Figure 2) is used to represent the blowing snow particle distribution function. There are three additional runs SI_Classic, SI_Area and SI_Mass (included in the prefix of experimental names) performed with aim of investigating possible mechanisms involved

in the SSA production (see section 3.3.1). The control run for open ocean sea spray is SI_Base_OO, following the scheme by Jaeglé et al. (2011).

Apart from the global modelling investigations, an idealized theoretical calculation of SSA production flux is made to compare with the sea spray flux under the same wind speed of 12 m s$^{-1}$, as discussed in section 5.

## 3.2 Parameterizations of SSA from SIZ

### 3.2.1 SSA flux

According to the scheme proposed in Yang et al. (2008; 2010), the SSA flux from blowing snow is proportional to bulk sublimation flux, $Q_s$ (kg m$^{-2}$ s$^{-1}$) and snow salinity $\varsigma$ (in units of psu, practical salinity unit, normally measured in grams of salt per kg sea water). Bulk sublimation flux $Q_s$ can be calculated following the approach of Déry and Yau (1999, 2001), when environmental factors, such as wind speed, $RH$ and air temperature etc. are given.

In order to demonstrate how to calculate SSA flux from bulk sublimation flux, here we simplify things by assuming (1) all snow particles have a uniform salinity $\varsigma$ (e.g. 0.06 psu, close to the median salinity from the field data, or 0.92 psu, close to the mean salinity), and (2) one blown-snow particle only produces one SSA after sublimation. This unit ratio (=1) assumption dictates a low bound of SSA number production.

Under the above assumption, the corresponding dry NaCl size, $d_{dry}$, for a snow particle with an initial diameter of $d_i$ and salinity

of $\varsigma$, can be derived as

$$d_{dry} = d_i \left( \frac{\varsigma \rho_{ice}}{1000 \rho_{NaCl}} \right)^{1/3} \tag{1}$$

Where $\rho_{ice}$ (917 kg m$^{-3}$) is density of ice and $\rho_{NaCl}$ (2160 kg m$^{-3}$) is density of NaCl. Note, the factor 1000 applied in equation (1) converts units of psu to kg salt per kg sea water.

At steady state, the SSA number production flux, $F_{SSA}(d_{dry})$ (particle m$^{-2}$ s$^{-1}$), should equal the snow particle loss rate via

sublimation, and the replenishment rate of supplied newly generated blowing snow particles, $F_{Snow}(d_i)$ (particle m$^{-2}$ s$^{-1}$). Thus, in a given snow size bin, with corresponding sublimation flux $Q_s(d_i)$, we have these two fluxes

$$F_{Snow}(d_i) = F_{SSA}(d_{dry}) = \frac{Q_s(d_i)}{M_{H_2O}(d_i)} \tag{2}$$

where $M_{H2O}(d_i)$ is water mass in a snow particle with diameter of $d_i$.

$$M_{H_2O}(d_i) = \frac{1}{6}\pi d_i^3 \rho_{ice} \tag{3}$$

For SSA mass flux (in kg NaCl m$^{-2}$ s$^{-1}$) at dry diameter $d_{dry}$, we have

$$Q_{SSA}(d_{dry}) = F_{SSA}(d_{dry}) M_{SSA}(d_{dry}) \tag{4}$$

where $M_{SSA}(d_{dry})$ is mass of SSA particle with size $d_{dry}$.

$$M_{SSA}(d_{dry}) = \frac{1}{6}\pi d_{dry}{}^3 \rho_{NaCl} \tag{5}$$

Incorporating above equations (1-3) and (5), equation 4 can be re-written as:

$$Q_{SSA}(d_{dry}) = Q_s(d_i)\frac{\varsigma}{1000} \tag{6}$$

which means NaCl mass flux is proportional to snow salinity and corresponding sublimation flux.

Obviously, how to derive $Q_s(d_i)$ for each snow size bin from the bulk sublimation flux $Q_s$ is key in the parameterization because it determines the size distribution of sea salt aerosol. As proposed, it needs two relevant parameters: (1) blowing snow size distribution function $f(d_i)$, and (2) snow particle mass loss rate, namely $\frac{dm_i}{dt}$, with $m_i$ the mass of a snow particle in size of $d_i$.

At steady state, when snow size distribution does not change with time, the combination term, $f(d_i)\frac{dm_i}{dt}$, could represent the

water loss rate for all-size particles. Unlike $f(d_i)$, $\frac{dm_i}{dt}$ is normally expressed in a non-normalized function, thus to allow a proper allocation, a normalization calculation for term $f(d_i)\frac{dm_i}{dt}$ is needed first. This can be done via a simple approach.

$$f_{norm}(d_i) = \frac{f(d_i)\frac{dm_i}{dt}}{\sum_{i=1}^{n} f(d_i)\frac{dm_i}{dt}} \tag{7}$$

where $n$ is the number of snow size bins. Note, at $\frac{dm_i}{dt}\propto$constant, $f_{norm}(d_i) = f(d_i)$.

With equation (7), the bulk sublimation flux can be allocated into each snow size bin.

$$Q_s(d_i) = Q_s f_{norm}(d_i) \tag{8}$$

Then SSA flux in equation (6) can be re-expressed as:

$$Q_{SSA}(d_{dry}) = Q_s f_{norm}(d_i)\frac{\varsigma}{1000} \tag{9}$$

It is likely that snow salinity is not constant in time, as the accumulated snow represents successive snowfalls and perhaps the influence of intermittent inputs from wind-blown sea spray and flooding. In this scenario, snow salinity is instead represented

by a frequency distribution, $\psi(\varsigma)$ and the integrated SSA production flux can be expressed as

$$Q_{SSA} = Q_s \iint f_{norm}(d_i)\psi(\varsigma)\frac{\varsigma}{1000} d(d_i)d\varsigma \tag{10}$$

Comparing to the equation (8) in Yang et al. (2008), we can find that the one in Yang et al. (2008) is a simplified version of the above equation at a condition of $\frac{dm_i}{dt}\propto$constant.

If more than one SSA is formed per snow particle, and assuming they are all equal in size, then at a ratio of $N$, the corresponding

dry NaCl size will be

$$d^*{}_{dry} = \left(\frac{1}{N}\right)^{1/3} d_i \left(\frac{\varsigma \rho_{ice}}{1000 \rho_{NaCl}}\right)^{1/3} \tag{11}$$

Under this condition, the SSA number flux will be simply $N$ times of the flux in equation (2) at $N=1$.

Figure 3 shows equivalent dry NaCl diameter (μm) as a function of initial snow particle diameter (μm) and snow salinity (psu)

calculated following equation (1) or (11) at N=1. For an initial snow particle $d_i$=10 μm, the SSAs formed at salinity <10 psu are sub-micron sized; at low salinity <0.01 psu, the SSAs are <0.1 μm. For a larger snow particle $d_i$ =100 μm (close to the median size of the blowing snow), the SSAs formed are mainly in range of 1-10 μm at salinity ranging 0.01-10 psu; at an even lower salinity of <0.01 psu, the SSAs are sub-micron sized. Note, at N=~10, the corresponding dry SSA size will be roughly half of the value at N=1.

### 3.2.2 Blowing snow particle flux

As pointed out above, at steady state, snow particle loss rate via the sublimation process should be balanced by newly supplied/generated blowing snow particles for each size bin to keep the snow particle size distribution unchanged with time. In windy conditions, vertical mixing via eddy turbulence is relatively fast, thus the time scale of mixing could be much shorter than that for the sublimation process. For instance, for a droplet with size of tens microns, to evaporate it completely may take a few thousands of seconds (Mason, 1971), which is substantially longer than the time scale of tens to hundreds of seconds in boundary layer turbulent mixing (Caughey et al., 1979). Therefore, the newly generated small snow crystals could be efficiently brought upwards, via rebound and splashing of snow grains in the saltation layer (<0.1 m), to replenish sublimated ones. Under the assumption that one blowing snow particle only forms one SSA, then equation (2) can be used to describe the flux of blowing snow particle production rate.

### 3.3 Parameters and model experiments

### 3.3.1 Sublimation function

As shown in Table 1, there are four sublimation functions applied to the $\frac{dm_i}{dt}$ term to deal with bulk sublimation allocation. All control runs (with SI_Base in the prefix) apply a function of $\frac{dm_i}{dt} \propto$ constant (across size bins) in equation (7). This water loss rate can be re-expressed as $\frac{dr}{dt} \propto \frac{1}{r^2}$, with $r$ representing radius of a spherical crystal of equivalent mass $m$. There are two possible physical processes that could cause this relationship. The first is the so-called Kelvin curvature effect (Pruppacher and Klett, 1997), in which vapour pressure is higher above a curved surface so that small particles evaporate faster than large ones, and indeed in some circumstances large particles may actually grow at the expense of small ones. The second is the so-called 'air ventilation' effect, a process that can accelerate particle sublimation rate in turbulent air. For example, in an air-flow tube experiment under sub-saturation, crystals in size ranging 0.3-1.3 mm show a linear water mass loss rate (against time) (Thorp and Mason, 1967) suggesting that smaller particles are losing mass at the same rate as larger ones.

In SI_Classic runs (with SI_Classic in the prefix), a simple relation function of $\frac{dm_i}{dt} \propto d_i$ (or equally $\frac{dr}{dt} \propto \frac{1}{r}$) is applied. This is an sublimation rate for particles at a stationary condition (e.g. not moving relative to the surrounding air), at which water loss rate is controlled by the moisture gradient between the particle surface and the ambient air (Houghton, 1933). As shown in section 4 and 5, SI_Classic runs allocate relatively less water to smaller snow size bins than SI_Base runs, and therefore produce fewer

sub-micron sized SSA (the break-up effect is not considered here).

A third sublimation function of $\frac{dm_i}{dt} \propto d_i{}^3$ (or $\frac{dr}{dt} \propto r$) is investigated (denoted as SI_Mass). Note that there is no actual micro-physical process within the blowing snow layer that can be assigned to match this function, but it would be the case if an air parcel, including blowing snow unsorted by particle size, came into contact with sub-saturated air so that the entire population

became aerosol. If this occurred, then the SSA size distribution would be the same as the suspended blowing snow particles (from a snapshot of the blowing snow layer) that lose water completely.

A fourth function of $\frac{dm_i}{dt} \propto d_i{}^2$ (or $\frac{dr}{dt} \propto$ constant) is also investigated (denoted as SI_Area). Again, we could not assign an actual micro-physical process to match it, but as it expresses in this function, it implies that the water loss rate is simply proportional to the particle surface area.

We hope by comparing the SSA size spectrum between model integrations and the observations, we may assess which of these functions could be most appropriate.

### 3.3.2 Blowing snow size distribution

It has been found that suspended blowing snow particles follow a two-parameter gamma distribution function $f(d_i, \alpha, \beta)$, with $\alpha$ shape parameter and $\beta$ scale parameter following a simple relationship of $\alpha\beta=D$, where $D$ is mean diameter in microns of

blown snow particles (e.g. Schmidt 1982; Dover 1993).

$$f(d_i, \alpha, \beta) = \frac{e^{-\frac{d_i}{\beta}} d_i{}^{\alpha-1}}{\beta^\alpha \Gamma(\alpha)} \tag{12}$$

The SPC instrument mounted at the Crow's nest showed a mean snow particle diameter of ~140 μm with $\alpha$ of 2~3 on average (Figure 2). The SPC instrument samples snow particles in the range of 46-500 μm, but due to the large background noise from the smallest (~46 μm) and largest (~500 μm) size bin, these two bins are not included in the data analysis and comparison.

Comparing to the snow data collected at Halley station, a coastal site in the Weddell Sea, which shows a similar $\alpha=2$ and mean diameter of ~150 μm (Mann et al., 2000).

The $\alpha$ value can vary from site to site and normally increases with increasing altitude from the surface (e.g. Nemoto and Nishimura, 2004). It is unlikely that $\alpha$ can be less than one, as $\alpha=1$ means the gamma distribution function will decay to an exponential function, which is not appropriate in describing blowing snow particles. Due to the lack of instrumental data at

size < 46 μm, we could not precisely describe the blowing snow size distribution function. For this reason, two distribution modes are applied (Figure 2): mode A having $\alpha=2$ with $\beta=70$ μm, and mode B having $\alpha=3$ with $\beta=46.7$ μm (both with a fixed mean diameter of 140 μm). Note that the two gamma functions (modes A and B) cannot be used to compare directly to the observed data (black line, Figure 2) because of different sampling size ranges used in their normalization calculations.

### 3.3.3 Snow salinity

Similar to the previous modelling study by Rhodes et al. (2017), a surface snow salinity distribution is applied (e.g. see Figure 12 of Frey, et al. 2019), which is based on surface snow samples (top 10 cm) collected in the Weddell Sea SIZ. In the Arctic, snow salinity values are trebled to reflect the likelihood that snow there is more saline than in the Antarctic due to reduced

precipitation rate (Yang et al., 2008). Further, we make the rate of SSA emission from multi-year sea ice half that from first-year sea ice (Rhodes et al., 2017). We note that these assumptions will not affect the main conclusions of this study.

As reflected in equation (1), SSA size is proportional to salinity with a power of 1/3, thus for a 10-fold change in salinity, dry NaCl size only alters by a factor of ~2. However, since snow salinity can vary by orders of magnitude, e.g. from the lowest values of $10^{-3}$ to more than 100 psu, snow salinity is an important factor in determining both SSA size and mass loading. To

test model sensitivity to this factor, two fixed snow salinity experiments are performed, with SI_Base_A_SL applying a fixed low value of 0.06 psu (median) and SI_Base_A_SH a high salinity of 0.92 psu (mean). We also include an experiment to test the sensitivity to highly saline snow samples, e.g. with salinity >10 psu, which account for ~4% of total snow samples as measured. SI_Base_A_SN is the same as SI_Base_A but without samples at salinity >10 psu (Table 1).

### 3.3.4 Snow age

How snow age affects blowing snow and SSA production is not completely clear, though it has generally been thought that aged snow will be more resistant to wind-mobilization (Li and Pomeroy, 1997; Box et al., 2004). Snow age was initially introduced to the parameterization to counteract the relatively high snow salinity used (Yang et al., 2008). At present, this parameter amounts to a crude tuning tool with no clear physical meaning. Snow age =0 gives the largest coefficient (=1) to the production flux, therefore, by setting snow age to zero, we effectively remove this parameter altogether. Note that 'snow age'

should not be interpreted as the time elapsed after the snowfall.

Actually, the 'snow' here refers to all ice crystals on the surface snow pack that can be up-lifted by air movement. These include fresh fallen snow, diamond-dust, wind-cropped frosts or even 'aged' snow that has been re-mobilized by wind-erosion. The mixing of fresh snow and 'old' saline snow changes the salinity distribution, a process that has not been considered by the model so far. Due to lack of data, we do not know how fast fresh fallen snow acquires salts. This process may be fast and

efficient during windy conditions through direct physical contact with salt-rich crystals. With further data, we may have a better representation of this process.

Here in this study, we follow a recent study by Huang and Jaeglé (2017) by setting a snow age =1.5 days for the southern hemisphere and 3 days for the northern hemisphere, which is slightly different to our previously value of 1 day in both hemispheres (Rhodes et al., 2017). This change causes reductions of ~16% in the southern hemisphere and ~39% in the

northern hemisphere in bulk sublimation flux.

### 3.3.5 Relative Humidity (*RH*)

As pointed out by Mann et al. (2000), sublimated water from blowing snow particles will raise the *RH* (with respect to ice) within the blowing snow layer, which will have a negative effect on the further sublimation of wind-blown snow particles, especially from the near surface layer. A model without consideration of this negative feedback may likely overestimate sublimation and SSA production. The p-TOMCAT model gets its *RH* field directly from ECMWF ERA-interim data. Therefore, it is likely that the model surface *RH* is underestimated in the cases with blowing snow. Figure 1c indicates that the lowest model gridbox *RHs* (w.r.t. ice) (at an average height of ~30 m) are, on average, significantly lower than the observations, which may be responsible for some overestimated SSA events by the model. To test model sensitivity, two runs are performed, with SI_Base_A_R1 applying a fixed surface *RH*(w.r.t. ice) =90% and SI_Base_A_R2 applying *RH*(w.r.t. ice) =95% (Table 1). SI_Base_A_R2 run result is shown in Figure 1a (orange line).

### 3.3.6 Threshold wind speed

According to Li and Pomeroy (1997), the threshold wind speed for blowing snow is air temperature and snow age dependent. According to the bulk sublimation parameterization of Déry and Yau (1999, 2001), a minimum threshold of ~7 ms$^{-1}$ is obtained at air temperature around -27°C. With the equation used, the threshold wind speed will be larger at either warmer or colder conditions. For example, at air temperatures of -10 and 0°C, as experienced during the cruise, the model calculates a threshold wind speed of ~8 and 10 ms$^{-1}$ respectively (Fig 1b). Note that the above equation may overlook blowing snow events at low wind speeds. For example, the onset of saltation or drifting snow can be observed at wind speed of just a few meters per second for loose dry and/or unbounded fresh snow (e.g. Male 1980; Pomeroy et al. 1993; King and Turner, 1997; Nemoto and Nishimura, 2004; Doorschot et al., 2004; Clifton et al., 2006). Since this process is not reflected by the model, therefore, it could explain those underestimated or completely missed SSA enhancement events, e.g. the aerosol spikes occurring during 11-13 July (Figure 1a).

Due to the large perturbation in air temperature, the threshold wind speed calculated varies significantly in association with the temperature perturbation (orange line in Figure 1b). To test model sensitivity to this term, model runs with fixed threshold speeds of 7, 8 and 9 m s$^{-1}$ (in SI_Base_A_T1, SI_Base_A_T2, and SI_Base_A_T3, respectively) (Table 1) are performed.

### 3.3.7 SSA production ratio per snow particle

In the original parameterizations (Yang et al., 2008), a unit ratio (N=1) is assumed, namely only one SSA is formed from one single snow particle. However, we cannot rule out the possibility that more than one SSA can be formed during the sublimation process, for example, by collision of snow particles in the saltation layer or the dynamical effect for snow particles in turbulent air. So far, this issue is quite unclear. A ratio of N=5 has been applied in a recent modelling study (Huang and Jaeglé, 2017) to allow a better agreement to the observations. Here we have two experiments, with N=10 in SI_Classic_AX10 and N=20 in

SI_Classic_BX20. Results are discussed in section 5 and shown Table 2. Note that the selection of N=10 or 20 is arbitrary and simply a model experimental trial.

## 4 Results of the model-data comparison

### 4.1 In the Weddell Sea

Figure 1a shows a comparison of observed total aerosol number density along the cruise track and model output (size ranging ~0.4-12 μm) from i) control run SI_Base_A, ii) a reduced surface relative humidity experiment SI_Base_A_R2 (using only size bins overlapping the instrumental size), iii) open ocean sea spray source. Model results clearly indicate that sea spray (green line) dominated aerosol signals before the vessel entered the sea-ice zone on 17 June; subsequently, sea-ice sourced SSA played the dominant role of generating aerosol when the vessel entered deep into the SIZ.

For the full analysis, we have re-grouped the cruise data into three surface types: open ocean (before 17 June), marginal sea-ice, and packed sea-ice, using air temperature of -18°C as the threshold between marginal ice and packed sea-ice. According to this classification, open ocean, marginal ice and packed sea-ice account for 9%, 42% and 49%, respectively, of the measurements. The corresponding mean air temperatures are -0.8, -11 and -22°C, with mean wind speeds of 9, 10.3 and 8 m s$^{-1}$, for each zone. Note a similar result can be obtained if the model's sea-ice (or open water) coverage fraction is used for the

classification (not shown).
    Figure 4 shows the simulated aerosol size spectrum in each surface zone. It can be seen that over the open ocean (Figure 4a), sea spray (OO, blue line) dominates over sea-ice sourced SSA (in three model runs SI_Base_A, SI_Classic_AX10 and SI_Classic_BX20). By looking at the time series, we find that sea spray shows a significant positive correlation to the observations with a correlation coefficient of r=0.55 (Table 2). The model-data ratio in Table 2 (for overlapping size range)

suggests that the model underestimates the observation by ~50% in the open ocean zone.
    Once the vessel enters densely packed sea-ice (Figure 4c), the simulated sea spray contribution drops significantly to only ~10% and cannot explain the observations. Meanwhile, sea-ice sourced SSA dominates, although the above three runs overestimate the observations with model-data ratios of 1.8~2.8. In addition, they all show a positive correlation to the observations with a coefficient r>0.5 (Table 2).

In marginal sea-ice (Figure 4b), our simulations suggest that both sea-ice and open ocean sourced SSA are making a contribution to the observations. However, neither sea-ice sourced nor sea spray alone shows a strong positive correlation with the observations. For example, the time series show only a small positive coefficient r=0.25 for all the three sea-ice sourced SSA with r=0.14 for sea spray. Their combined effect (sum of sea-ice sourced SSA and sea spray) only shows a slight increase in the relationship coefficient with r of 0.27-0.28 (Table 2), indicating limited model ability in marginal ice SSA simulation.

In the marginal ice zone, the model (sum of sea-ice and open ocean sourced SSA) underestimates the observations by ~30%, as shown in the third column of Table 2, where a ratio of 0.19 for sea spray (OO) and 0.47 for sea ice sourced SSA (SI_Base_A) is obtained respectively. . The lack of significant correlation in the marginal zone could be related to the large variations in air

temperature and moisture in both temporal and spatial scales in this transitional surface zone. Moreover, since the parameterization for bulk sublimation flux calculation was derived based on data at relatively colder and drier conditions, e.g. from the Canadian Prairies, whether it is applicable in warmer conditions, such as over sea ice, is not yet known.

Although the meteorological fields, such as wind speed (Figure 1b), temperature (Figure 1c) and moisture (Figure 1d), taken from the ERA-interim database, in general agree well to the observations, discrepancies between them can be large during specific time periods. On average, model surface wind speeds are lower than the observations, especially during storm events; this is because global models with a coarse spatial resolution suffers significant spatial averaging and cannot reproduce gusty winds. For example, a mean wind speed of ~22 m s$^{-1}$ is observed during UTC 12:00 27$^{th}$ and UTC 6:00 28$^{th}$ June, which is ~7 m s$^{-1}$ higher than the lowest model layer wind speed (at ~30m). This lower model wind speed means an underestimation in both sublimation and SSA by a factor of ~2. Given that the sublimation flux from blowing snow is a function of wind speed with a power of ~3, then the largest model underestimation for SSA production are likely in association with storm events.

At air temperatures of -35 to -20°C, the threshold wind speed for blowing snow (calculated from the formula (2) in Yang et al. 2008) stays low, e.g. ~7 m s$^{-1}$; however, it increases to ~8 m s$^{-1}$ at a temperature of -10°C and ~10 m s$^{-1}$ at just below zero. At the marginal sea-ice zone, air temperature suffers large perturbations, making threshold wind speed very variable (Figure 1c), affecting both sublimation and then SSA production calculations. It is interesting to note that model runs with fixed threshold speed (7 m s$^{-1}$ in SI_Base_A_T1 and 8 m s$^{-1}$ in SI_Base_A_T2) show better agreement with the observations in the marginal ice zone, with correlation coefficients increased from the control run r=0.25 to ~0.3 in those two runs (Table 2). The combined results from sea spray and sea-ice sourced SSA show a similar result, e.g. from r=0.28 (in OO + SI_Base_A) to 0.30 (in OO + SI_Base_A_T1) and 0.32 (in OO + SI_Base_A_T2). As a result, the SSA number densities in SI_Base_T1 shows an increase of ~50% over both the marginal and packed sea-ice zones. In SI_Base_A_T2 run, the concentrations drops by 40-50%. At an even higher threshold of 9 m s$^{-1}$ (in SI_Base_A_T3), the SSA production from blowing snow is strongly suppressed (Table 2).

During 11-13 July, there are two large aerosol enhancement events which are completely overlooked by the model. As shown in Figure 1b, they correspond to relatively low wind speeds (in both reality and model), lower than the calculated threshold speed of 7 m s$^{-1}$. However, as discussed in section 3.3.6, drifting snow can be measured at low wind speeds of just a few meter per second when snow particles are loose and unbounded, a process which has not been considered by the model. This possibly explains why the model fails to reproduce these two aerosol spiking events.

Apart from wind, moisture is another key factor that influences both sublimation and SSA production. As shown in Figure 1d, model *RHs* are generally lower than the observations, which is likely due to the model not considering the negative feedback of sublimated water vapour to the near surface blowing snow layer, which will prevent further water loss from suspended snow particles. Obviously, models without considering this feedback effect could result in overestimation of SSA flux at some circumstances. We perform two model experiments with fixed surface *RH*(w.r.t. ice) =90% in SI_Base_A_R1 and =95% in SI_Base_A_R2 to investigate this issue. As reflected in Figure 1a (orange line, with *RH*(w.r.t. ice) =95%) and Table 2, these two models results are much closer to the observations. For instance, the model-data ratio of aerosol number density in the

sea-ice zone reduces from the control run 2.76 to 1.8 in the SI_Base_A_R1 and 1.1 in the SI_Base_A_R2 (Table 2). As a result, the time series correlation coefficients between the model and the observations increase from r=0.56 in the control run (OO+SI_Base_A) to 0.64 in both OO + SI_Base_A_R1 and OO + SI_Base_A_R2 runs (Table 2).

Blowing snow particle size distribution function also affects SSA size distribution. A smaller $\alpha$ means there are more small snow particles (e.g. < tens of microns) in the distribution than a larger $\alpha$. Thus, model runs with mode A ($\alpha$=2) implemented normally produce more submicron sized SSA than with mode B ($\alpha$=3), as shown in Figure 5a.

When a SSA production ratio greater than 1 is applied, the size of the dry NaCl formed will be reduced (refer to equation 11). Thus, at N>1, the SSA spectrum will shift towards smaller size bins. Figure 5a shows that, in size range > 0.4 μm, Classic_AX10 and SI_Classic_BX20 give very similar result to SI_Base_A run, although with significant differences in smaller size bins (reflecting shape value effects, i.e. $\alpha$=2 in mode A vs $\alpha$=3 in mode B). Salinity not only affects salt mass loading, but also the size distribution of SSA generated. As mentioned before, highly saline snow samples (e.g. with salinity ≥ 10 psu) only account for a small fraction of measurements (e.g. ~4% of the Weddell Sea measurements). Model run (SI_Base_A_SN) without these saline snow samples shows a reduction of SSA concentration by >50% at SSA size of ~10 μm, and ~20% at submicron size mode (Figure 5b). Given that large aerosols dominate the salt mass budget, high salinity snow samples are important regarding total amount of mass loading and chemical compound release (such as bromine) in the boundary layer. In a run (SI_Base_A_SL) with a fixed low snow salinity of 0.06 psu (close to median salinity), a >90% reduction in SSA number density at size of ~10 μm (compared to the control run) is seen, with ~20% reduction at size < 0.1 μm. On the contrary, in a run (SI_Base_A_SH) with a fixed high salinity of 0.92 psu (close to mean salinity), an increase of ~100% in submicron SSA (compared to the control run) is seen. The above experiments indicate that snow salinity is an important factor in determining both SSA mass loading and size distribution. Geographically, the difference in snow salinity on sea ice is expected to be large. For example, large differences can be expected between the northern hemisphere vs southern hemisphere, young sea ice vs multi-year sea ice, marginal sea ice vs packed sea ice, etc. Even in the same geographic location, there could be seasonal evolution of snow salinity in associating with e.g. salt loading and precipitation dilution, etc. Currently we do not have a systematic measurement of snow salinity globally, which significantly impedes modelling efforts to simulate realistic representation of SSA and bromine chemistry (e.g. Huang and Jaeglé, 2017; Rhodes et al., 2017; Legrand et al., 2017).

## 4.2 Global scale

Global model studies show that the observed winter SSA mass peaks at most polar sites can only be reproduced when the sea-ice sourced SSA are implemented (Levine et al., 2014; Huang and Jaeglé, 2017; Rhodes et al, 2017). Figure 6 shows an updated p-TOMCAT result of seasonal [Na] concentrations at eight polar stations (based on a three-year integration 2013-2015), which reinforces the importance of sea-ice sourced SSA in reproducing the winter peaks of sodium observed, as sea spray (solid green lines) simply cannot do alone. As shown in Figure 6, model run SI_Classic_BX20 (solid yellow lines) gives a slightly higher [Na] mass concentrations than the control run SI_Base_A (red lines), this is due to the reduction of SSA size, e.g. by a factor of 2.7 when N=20 is applied. Model run SI_Base_A_R1 (solid purple lines, with a fixed *RH*=90%) gives slightly reduced

SSA mass concentrations comparing to SI_Base_A, but still shows a clear winter SSA mass peak in most polar sites. However, SSA mass in SI_Base_A_R2 (dotted purple lines, with a fixed $RH$=95%) is much suppressed and cannot represent the observations. Among all the sea ice sourced schemes, SI_Classic_A (dotted green lines lines) gives the least SSA mass and could not explain the winter peaks, which is due to the least sub-micron sized SSA being formed in SI_Classic_A.

The three model runs (SI_Base_A, SI_Classic_AX10 and SI_Classic_BX20) give very similar mass loading (Figure 6) and number density at size >0.4 μm, however, they are quite different in terms of number density at smaller size bins, especially in ultra-fine mode (Figure 5a). Figure 7 shows a zonal mean SSA number density from one year integration (2013) from these three runs. It can be seen that SI_Base_A has the largest SSA number production, with SSA number density over sea-ice higher than that of sea spray in the marine boundary layer (Figure 7a and d). On the contrary, SI_Classic_AX10 and SI_Classic_BX20

give much lower SSA number density with a maximum boundary layer density of ~5 and ~3 particles per cubic centimetre, respectively, still higher than the simulated sea spray contribution in winter season in polar regions.

With detailed blowing snow data, the shape parameter as well as the scale parameter can be well constrained, then the larger differences in predicted SSA number density in submicron mode among these model runs (mainly between SI_Base_A and SI_Classic_AX10 or SI_Classic_BX20) can be used as indicators for validation, when aerosol data in ultra-fine mode becomes

available from SIZ locations.

Overall, the control run SI_Base_A overestimates SSA number density (as shown in Figure 1a) and underestimates mass concentration at sites such as Alert, Barrow and Neumayer (Figure 6), indicating the current model setups and parameterizations applied need further constraints and evaluation against data. Model runs with a fixed $RH$=90% (in SI_Base_A_R2) seem to outperform the control run SI_Base_A indicating model moisture field, which determines the

sublimation flux calculation, is a crucial factor that may greatly affect simulations. In addition, inclusion of drifting snow (which is missed by the model) as a source of SSA may improve SSA mass simulation in polar regions, but will increase number density as well, thus still will not reconcile the discrepancy between the model (the control run) and the observations in both number density and mass concentration. As discussed previously, apart from the sublimation rate applied, blowing snow size distribution (shape parameter $\alpha$ and scale parameter beta $\beta$) can also affect SSA size spectrum, as shown in Figure

5a (e.g. SI_Base_A vs SI_Base_B). Cruise data show that blowing snow particle size distribution varies as a function of height above the surface and wind speed (see details in Frey et al., 2019). Therefore it is important to apply a more realistic blowing snow distribution to constrain this key parameter; we plan to investigate this issue by applying a time-series of observed blowing snow size distribution along the cruise track to further constrain this parameter to narrow down the uncertainty.

**5 Possible physical mechanisms involved in the SSA production from blowing snow**

As discussed in section 3.3.1, under the assumption that one snow particle only forms one SSA after sublimation, SI_Mass_A run shows the least correspondence to the cruise observations, by several orders of magnitude (Figure 5a). Thus, it is safe to rule out the physical mechanism represented by the sublimation function implemented, which assumes that the SSA should

come from an unsorted sample of suspended wind-blown snow particles in the blowing snow layer that lose their water completely without any replenishment from newly generated snow particles. SI_Classic_A and SI_Classic_B runs agree better than SI_Area_A and SI_Mass_A runs, but still cannot generate enough submicron size SSA to match the observations. SI_Base_A and SI_Base_B are, instead, much closer to the observations with mode-data ratios ranging within 0.8~2.8 (Table

2). As discussed previously, SI_Base runs claim a particle sublimation function of $\frac{dm_i}{dt}\propto$constant or $\frac{dr}{dt}\propto\frac{1}{r^2}$, demanding the water loss is dominated by the curvature effect and/or the so-called air ventilation effect. Instead, SI_Classic run applies a well-known function of $\frac{dr}{dt}\propto\frac{1}{r}$, indicating the water loss is controlled by the moisture gradient (diffusion effect) for a snow particle in sub-saturated air.

There is a possibility that more than one SSA could be formed from one saline snow particle. If this is the case, then the

discrepancies between SI_Classic_A (or SI_Classic_B) and the observations could be reduced. For example, when a SSA production ratio of 10 per snow particle is applied to SI_Classic_A (denoted as SI_Classic_AX10), or a ratio of 20 to SI_Classic_B (denoted as SI_Classic_BX20), then a result similar to the control run (SI_Base_A) in particle size of ~0.-12 µm can be obtained (Figure 1e or Figure 5a). For SI_Area_A a ratio of N=~100 is needed, with an even larger ratio needed for SI_Mass_A, to match the observations (not shown). However, the current cruise data will not allow us to separate or pin-point

which process is more plausible, demanding further investigation on this issue.

Cruise data show that blowing snow particle number densities decrease significantly, e.g. by more than an order of magnitude from near surface (~2m above snow surface) to ~29 m. However, aerosol number densities between these two levels do not show such a large gradient as observed for blowing snow. For example, observed data indicate (see Figure 5 in Frey et al., 2019) that during drifting snow episodes aerosol number densities increased significantly especially of sub-micron sized

particles at both measurement heights, with a lightly greater increase near the surface (number density up to $10^7$ m$^{-3}$ for diameter <2µm). During blowing snow number densities showed similar increases as during drifting snow, however at 29 m concentrations were higher and particles were larger (at diameter >9 µm) than at 2 m. This observational evidence will not allow us to derive any robust conclusion regarding where SSA is generated: in the near surface layer where $RH$ is saturated or at the top of the blowing snow layer where $RH$ is likely under saturated. If SSA is mainly produced near the surface layer, then

snow particle sublimation will be controlled by the 'curvature effect' (following the SI_Base mechanism). However, if SSA is produced in the sub-saturated condition, e.g. at the top layer or above the blowing snow layer, then water sublimation will follow the SI_Classic mechanism.

Model experiments with the above two mechanisms implemented (e.g. SI_Base_A and SI_Classic_AX10) can produce roughly the same number density at size range of ~0.4-12 µm. However, at SSA size <0.4 µm diameter, their results are quite

different as shown in Figure 4c and Figure 5a. For example, at diameter of 0.1 µm, SI_Base_A has a mean SSA number density almost an order of magnitude larger than that of SI_Classic_AX10 (and SI_Classic_BX20). Therefore, a precise observation of SSA at sub-micron size mode can help to diagnose which micro-physical mechanism(s) dominates the SSA production. A systematic measurement of the size segregated chemical composition of SSA over a size range of 0.03 to 20 µm diameter,

together with a complete spectrum of blowing snow particle size will help to distinguish which mechanism dominates SSA production from blowing snow.

To highlight the above mentioned SSA production mechanisms and make a direct comparison with sea spray flux, a theoretical calculation is performed with results shown in Figure 8. The bulk sublimation flux is calculated under polar weather conditions
of wind speed=12 m s$^{-1}$, temperature=-10°C, and *RH* (w.r.t. ice)=80%, with a zero snow age and a constant snow salinity of 0.06 psu. It can be seen from Figure 8a that SI_Base_Aa allocates most water, higher than SI_Classic_Aa, to small snow particles at diameter < tens microns; while both SI_Area_Aa and SI_Mass_Aa allocate little water to these snow size bins. As a consequence, they have the smallest SSA production rate at submicron size mode and highest rate at micron size mode (Figure 8c, d).

At sub- to micron size mode, SI_Classic_Aa shows a comparable flux to sea spray calculated at the same wind speed following Jaeglé et al (2011) scheme with a SST=5°C (black dotted line in Figure 8) and Caffrey et al. (2006) scheme (black solid line). SI_Base_Aa and SI_Classic_AaX10 both show stronger SSA production flux at size of less than a few microns. At SSA diameter of 0.1-1μm, they both show a flux of >10 times that of sea spray (by OO_Jaeglé); at ultra-fine mode (<0.1 μm), SI_Base_A has a production flux larger than OO_Jaeglé flux by >2 orders of magnitude.

Apart from a nearly 10 times increase in the number density, compared to Classic_Aa, Classic_AaX10 also shows a shift of the SSA size spectrum towards smaller bins with a roughly halved NaCl size according to equation (11), indicating more smaller SSA formed, as shown in Figure 8d. Figure 8e shows that SI_Base_Aa and SI_Classic_AaX10 have the largest submicron SSA accumulation fraction, accounting for ~2% of the total mass, which is >10 times the sea spray fraction. This enhanced submicron size partitioning from the sea-ice surface may contribute to the observed enhancement of submicron size
SSA in polar winter (e.g. Rankin and Wolff et al. 2003; Quinn et al., 2002). Figure 8d also shows that at large SSA size, e.g. > 10 μm, blowing snow generates fewer SSA than sea spray, strongly indicating that sea spray and sea-ice sourced SSA have quite different size spectrum as their fingerprints.

The assumption that one blowing snow particle only forms one SSA after sublimation means, at steady state, that the SSA number production rate should be the same as the snow particle loss rate and the replenishment rate of newly formed snow
particle. For that reason, equation (2) can be used to describe blowing snow particle production flux (in vertical dimension) due to sublimation effect (Figure 8b). However, our cruise data will not allow us to validate this flux and derive any robust conclusion.

## 6 Conclusion

The Weddell Sea cruise data gives us a unique opportunity to constrain some key parameters involved to SSA production,
validate parameterizations and investigate possible micro-physical processes involved. Unfortunately, due to lack of data at smaller particle sizes, e.g. < 0.4 μm, we could not pin-point the exact mechanism that is responsible for SSA production from blowing snow. However, the current data and model integrations suggest two plausible mechanisms. The first one is under an

assumption that only one SSA is formed per snow particle. Under this assumption, to match the observations (in size ranging 0.4-12 µm), it demands that the curvature effect dominates water sublimation (as proposed in Yang et al. 2008). This mechanism implies that SSA should be generated under a saturated environment, e.g. near the surface layer, rather than in a layer on top of the blowing snow layer where sub-saturation is likely. The second mechanism allows for more than one SSA

formed per snow particle, due to the breaking-up effect. To match the observations, it demands a ratio of 10 SSA per snow particle for SI_Classic_A and a ratio of 20 for SI_Classic_B. This mechanism is built on the micro-physical process that snow sublimation rate is dominated by the moisture gradient between the snow surface and the ambient air (or the moisture diffusion effect). Although the ratio value needed (to match the observations) varies among different model setups (e.g. total sublimation flux and blowing snow size distribution), it clearly demands that SSA should be produced in a sub-saturated layer, e.g. on top

of the blowing snow layer, rather than inside of the blowing snow layer. However, the aerosol concentration gradient observed between near surface (~2m above snow surface) and ~29 m will not allow us to conclude robustly where the SSA is produced. In addition, the large biases in the converted *RH* (w.r.t. ice) (Frey et al. 2019) at the two heights also prevent us from pin-pointing which process is underlying the SSA production. Also, there is little knowledge regarding how air ventilation effect on crystal particle sublimation process, which may accelerate water vapour sublimation rate from all sizes of snow particles

and under both saturated and sub-saturated conditions. Thus, this highlights the need for further in-situ observations, laboratory investigation and modelling to fill this gap. Climate models are then critically needed to estimate the impact of this newly identified sea-ice sourced SSA on local and regional climate, directly (via scattering sunlight) and indirectly (via acting as cloud condensation nuclei, influencing cloud and precipitation).

**Author Contributions**

EW, MF, AJ and PA designed the field experiment, MF carried out the field measurements. XY designed and performed model experiments, interpreted model output and the micro-physical mechanisms proposed. RR contributed to model development, SN and IB to CLASP and KN to SPC instruments. XY prepared the manuscript with contributions from all co-authors.

**Acknowledgements**

We thank the Alfred Wegener Institute for Polar and Marine research, which made our participation in the cruise possible. We

gratefully acknowledge financial support from NERC/UK through the project BLOWSEA (NE/J023051/1 and NE/J020303/1). RHR was supported by a European Commission Horizon 2020 Marie Sklodowska-Curie Individual Fellowship (No. 658120, SEADOG). EW is supported by a Royal Society Professorship (RP120096 and RP/R/180003). XY thanks W. Feng and M. Chipperfield for support in using ERA-interim data. Thanks also go to S. Palm for useful discussion.

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

Table 1: Model experiments for sea ice sourced SSA (with SI in prefix of each experiment) and sea spray fluxes (with OO in the prefix). Columns 2-10 show parameters applied to each experiment: sublimation function, shape of blowing snow size distribution, ratio of SSA formed per blowing snow particle, snow age, salinity, threshold wind speed, $RH$ (w.r.t ice) and air temperature.

| Model experiments | $dm_i/dt$ (sublimation rate as a function of diameter $d_i$) | Shape $\alpha$ ($\alpha \times \beta = 140$ $\mu$m, mean diameter) | Ratio of SSA per blowing snow particle | Snow age (day) | Snow salinity (psu) | Threshold wind speed (m s$^{-1}$) | Surface wind speed (m s$^{-1}$) | $RH$ (%) w.r.t. ice | Surface temperature (°C) |
|---|---|---|---|---|---|---|---|---|---|
| SI_Base_A | constant | 2 | 1 | 1.5 | full distribution | calculated | ERA-interim | ERA-interim | ERA-interim |
| SI_Base_A_R1 | constant | 2 | 1 | 1.5 | full distribution | calculated | ERA-interim | 90 | ERA-interim |
| SI_Base_A_R2 | constant | 2 | 1 | 1.5 | full distribution | calculated | ERA-interim | 95 | ERA-interim |
| SI_Base_A_T1 | constant | 2 | 1 | 1.5 | full distribution | 7 | ERA-interim | ERA-interim | ERA-interim |
| SI_Base_A_T2 | constant | 2 | 1 | 1.5 | full distribution | 8 | ERA-interim | ERA-interim | ERA-interim |
| SI_Base_A_T3 | constant | 2 | 1 | 1.5 | full distribution | 9 | ERA-interim | ERA-interim | ERA-interim |
| SI_Base_A_SN | constant | 2 | 1 | 1.5 | distribution, without >10psu | calculated | ERA-interim | ERA-interim | ERA-interim |
| SI_Base_A_SH | constant | 2 | 1 | 1.5 | 0.92 | calculated | ERA-interim | ERA-interim | ERA-interim |
| SI_Base_A_SL | constant | 2 | 1 | 1.5 | 0.06 | calculated | ERA-interim | ERA-interim | ERA-interim |
| SI_Base_B | constant | 3 | 1 | 1.5 | full distribution | calculated | ERA-interim | ERA-interim | ERA-interim |
| SI_Classic_A | $d_i$ | 2 | 1 | 1.5 | full distribution | calculated | ERA-interim | ERA-interim | ERA-interim |
| SI_Classic_AX10 | $d_i$ | 2 | 10 | 1.5 | full distribution | calculated | ERA-interim | ERA-interim | ERA-interim |

| | | | | | | | | | |
|---|---|---|---|---|---|---|---|---|---|
| SI_Classic_B | $d_i$ | 3 | 1 | 1.5 | full distribution | calculated | ERA-interim | ERA-interim | ERA-interim |
| SI_Classic_BX20 | $d_i$ | 3 | 20 | 1.5 | full distribution | calculated | ERA-interim | ERA-interim | ERA-interim |
| SI_Mass_A | $d_i^3$ | 2 | 1 | 1.5 | full distribution | calculated | ERA-interim | ERA-interim | ERA-interim |
| SI_Area_A | $d_i^2$ | 2 | 1 | 1.5 | full distribution | calculated | ERA-interim | ERA-interim | ERA-interim |
| SI_Base_Aa | constant | 2 | 1 | 0 | 0.06 | calculated | 12 | 80 | -10 |
| SI_Classic_Aa | $d_i$ | 2 | 1 | 0 | 0.06 | calculated | 12 | 80 | -10 |
| SI_Classic_AaX10 | $d_i$ | 2 | 10 | 0 | 0.06 | calculated | 12 | 80 | -10 |
| SI_Mass_Aa | $d_i^3$ | 2 | 1 | 0 | 0.06 | calculated | 12 | 80 | -10 |
| SI_Area_Aa | $d_i^2$ | 2 | 1 | 0 | 0.06 | calculated | 12 | 80 | -10 |
| OO | N/A | N/A | 1 | N/A | N/A | calculated | ERA-interim | N/A | N/A |
| OO_Jaeglé | N/A | N/A | 1 | N/A | N/A | calculated | 12 | N/A | N/A |
| OO_Caffrey | N/A | N/A | 1 | N/A | N/A | calculated | 12 | N/A | N/A |

Table 2: Ratios of aerosol number density between model runs and the observations (for overlapping size range of 0.4-12 µm) along the cruise track over surface type of open ocean (column 2), marginal sea ice (column 3) and packed sea ice (column 4). The values in brackets are correlation coefficient between time series of model output and the observation at each surface zone.

| Experiments | Ratio (Model/Obs) over open ocean | Ratio (Model/Obs) over marginal ice | Ratio (Model/Obs) over packed ice |
|---|---|---|---|
| OO | 0.50 (r=0.55) | 0.19 (r=0.14) | 0.10 (r=0.33) |
| SI_Base_A | <0.01 (r=0.14) | 0.47 (r=0.25) | 2.76 (r=0.55) |
| OO + SI_Base_A | 0.50 (r=0.55) | 0.66 (r=0.28) | 2.86 (r=0.56) |
| SI_Base_A_R1 | <0.01 (r=0.14) | 0.45 (r=0.26) | 1.82 (r=0.63) |
| OO + SI_Base_A_R1 | 0.50 (r=0.55) | 0.64 (r=0.27) | 1.92 (r=0.64) |
| SI_Base_A_R2 | <0.01 (r=0.14) | 0.27 (r=0.26) | 1.10 (r=0.62) |
| OO + SI_Base_A_R2 | 0.49 (r=0.55) | 0.45 (r=0.26) | 1.21 (r=0.64) |
| SI_Base_A_T1 | 0.02 (r=0.14) | 0.98 (r=0.29) | 3.57 (r=0.58) |
| OO + SI_Base_A_T1 | 0.50 (r=0.55) | 1.17 (r=0.30) | 3.67 (r=0.58) |
| SI_Base_A_T2 | <0.01 (r=0.14) | 0.43 (r=0.33) | 1.27 (r=0.53) |
| OO + SI_Base_A_T2 | 0.50 (r=0.55) | 0.62 (r=0.32) | 1.38 (r=0.56) |
| SI_Base_A_T3 | <0.01 (r=0.14) | 0.22 (r=0.34) | 0.53 (r=0.50) |
| OO + SI_Base_A_T3 | 0.50 (r=0.55) | 0.40 (r=0.30) | 0.63 (r=0.54) |
| SI_Base_A_SN | <0.01 (r=0.14) | 0.38 (r=0.26) | 2.16 (r=0.55) |
| OO + SI_Base_A_SN | 0.50 (r=0.55) | 0.56 (r=0.28) | 2.27 (r=0.56) |
| SI_Base_A_SL | <0.01 (r=0.14) | 0.32 (r=0.26) | 1.82 (r=0.55) |
| OO + SI_Base_A_SL | 0.50 (r=0.55) | 0.50 (r=0.28) | 1.92 (r=0.56) |
| SI_Base_A_SH | 0.01 (r=0.14) | 0.94 (r=0.25) | 5.46 (r=0.54) |
| OO + SI_Base_A_SH | 0.51 (r=0.55) | 1.13 (r=0.27) | 5.57 (r=0.55) |
| SI_Base_B | <0.01 (r=0.14) | 0.13 (r=0.25) | 0.79 (r=0.54) |
| OO + SI_Base_B | 0.50 (r=0.55) | 0.32 (r=0.24) | 0.89 (r=0.57) |
| SI_Classic_A | <0.01 (r=0.14) | 0.05 (r=0.25) | 0.28 (r=0.53) |
| OO + SI_Classic_A | 0.50 (r=0.55) | 0.23 (r=0.19) | 0.39 (r=0.59) |
| SI_Classic_AX10 | <0.01 (r=0.14) | 0.32 (r=0.25) | 1.85 (r=0.53) |
| OO+ SI_Classic_AX10 | 0.50 (r=0.55) | 0.50 (r=0.27) | 1.96 (r=0.59) |
| SI_Classic_B | <0.01 (r=0.14) | 0.02 (r=0.25) | 0.15 (r=0.52) |
| OO + SI_Classic_B | 0.50 (r=0.55) | 0.21 (r=0.17) | 0.25 (r=0.57) |
| SI_Classic_BX20 | <0.01 (r=0.14) | 0.40 (r=0.25) | 2.38 (r=0.54) |
| OO+ SI_Classic_BX20 | 0.50 (r=0.55) | 0.50 (r=0.27) | 2.48 (r=0.55) |
| SI_Area_A | <0.01 (r=0.14) | 0.06 (r=0.25) | 0.04 (r=0. 52) |

| OO + SI_Area_A | 0.50 (r=0.55) | 0.19 (r=0.15) | 0.15 (r=0.46) |
| SI_Mass_A | <0.01 (r=0.14) | <0.01 (r=0.25) | <0.01 (r=0.49 ) |
| OO + SI_Mass_A | 0.50 (r=0.55) | 0.18 (r=0.13) | 0.11 (r=0.36 ) |

Figure 1: (a) time series of total aerosol number densities from observations along the cruise track (refer to Figure 1 in Frey et al. 2019) and model output of SI_Base_A (control run) and SI_Base_R2 (with a fixed surface *RH* (w.r.t. ice)=95%). Note, only SSA with sizes overlapping with the observation (~0.4-12 µm) are counted. Meteorological data of wind speeds (b), temperatures (c) and relative humidity with respect to ice from both observation and model are shown. Calculated threshold wind speed for blowing snow is given in (b). Penal (e) is same as

5    (a) apart from model output of SI_Classic_AX10 and SI_Classic_BX20.

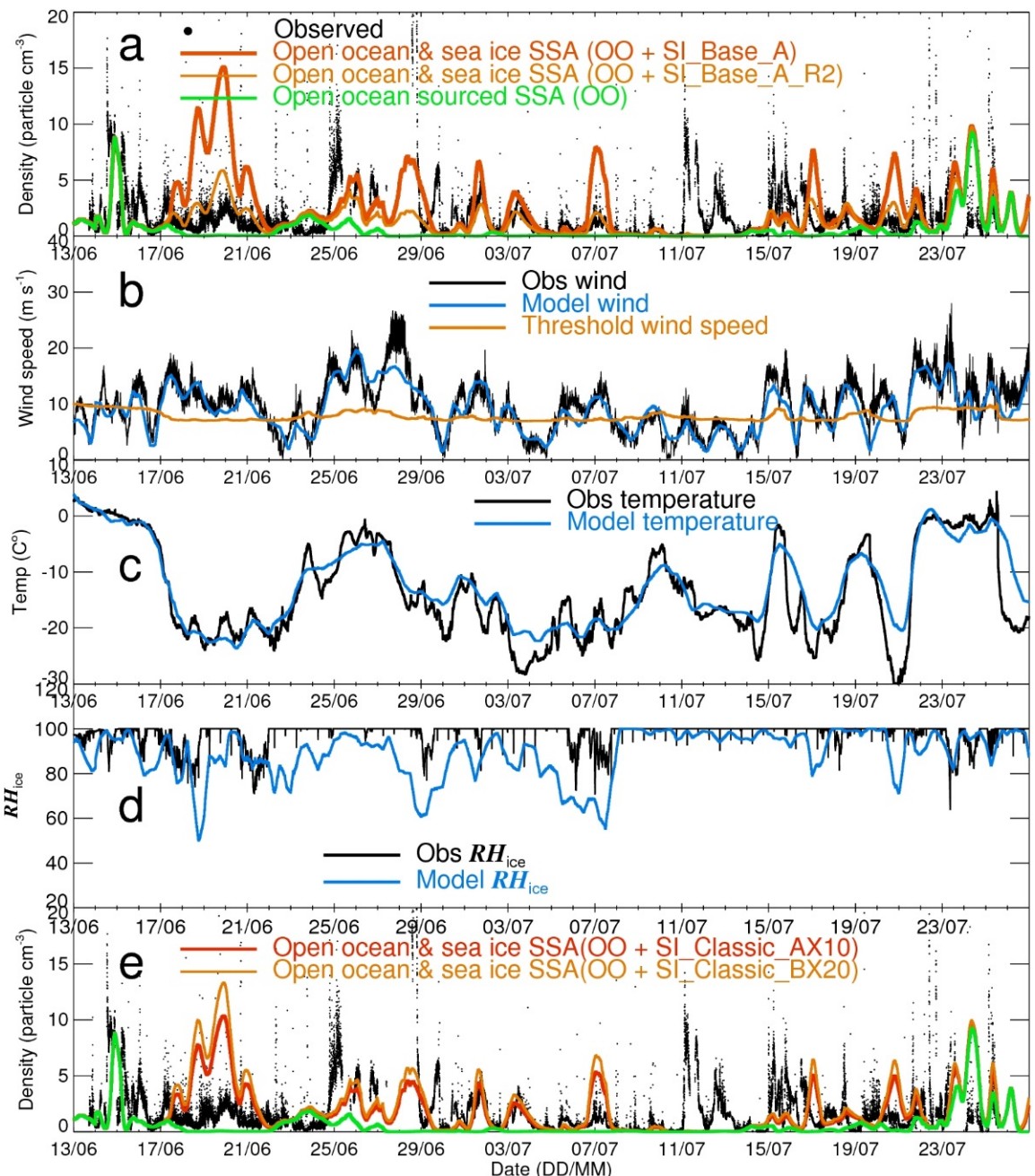

Figure 2: Normalized 29m SPC instrumental blowing snow size distribution is shown in black line. Note, dotted line is for small particles with diameter < 60μm. Two blowing snow size distribution functions are derived for model usage with mode A (red line) having a shape value $\alpha=2$ with $\beta$ =70μm, and mode B (blue line) having $\alpha=3$ with $\beta$ =46.7μm (fixed mean diameter=140 μm) . The X-axis interval is 10 μm.

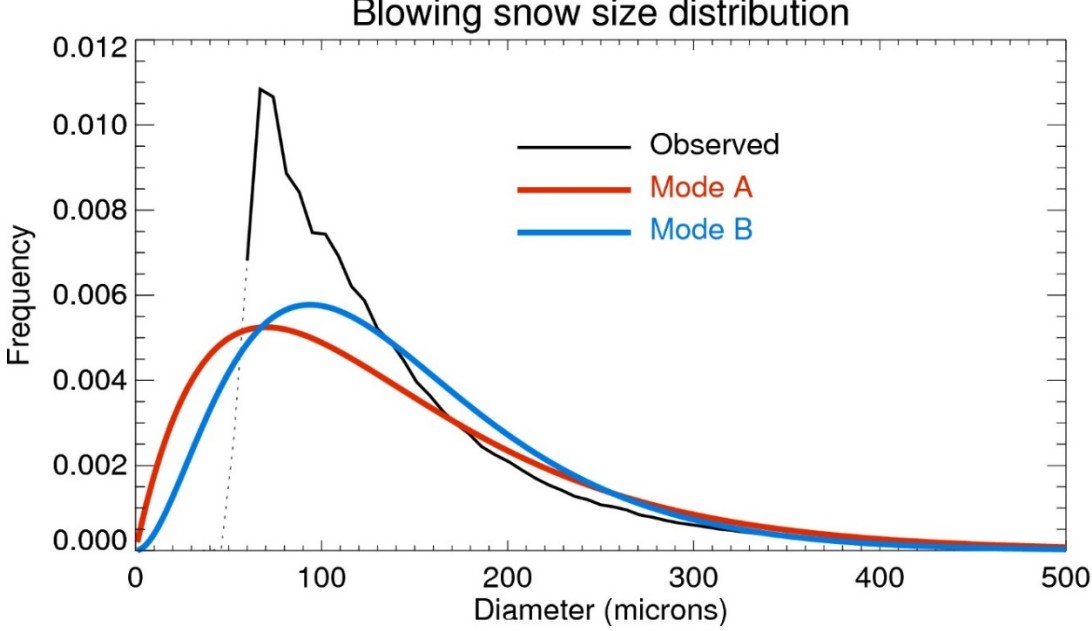

Figure 3: Equivalent dry SSA diameter (μm) as a function of initial snow particle diameter (μm) and snow salinity (psu) at N=1, calculated following equation (11).

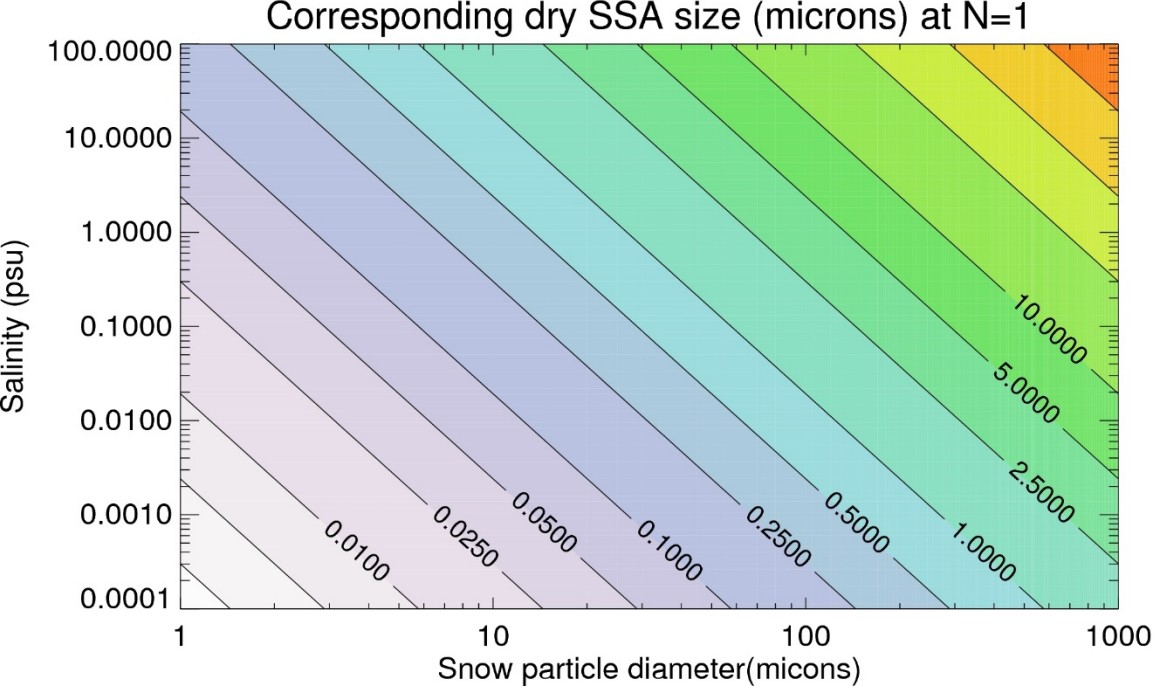

Figure 4 Size distribution of sea spray and sea ice soured SSA at three defined surface zones: (a) open ocean, (b) marginal sea ice, and (c) packed sea ice zone. Observations are shown by the black lines with box symbols. Open ocean sea spray comes from the OO run (blue). Sea ice sourced SSA from the control run SI_Base_A (red) and two SI_Classic runs. Note, SI_Classic_AX10 (green) is same as SI_Classic_A but applying a ratio of 10 SSA produced per blowing snow particle; SI_Classic_BX20 (orange) is same as SI_Classic_B but applying a ratio of 20 (Table 1). Vertical dashed lines represent diameter of 0.1, 10 and 10 µm.

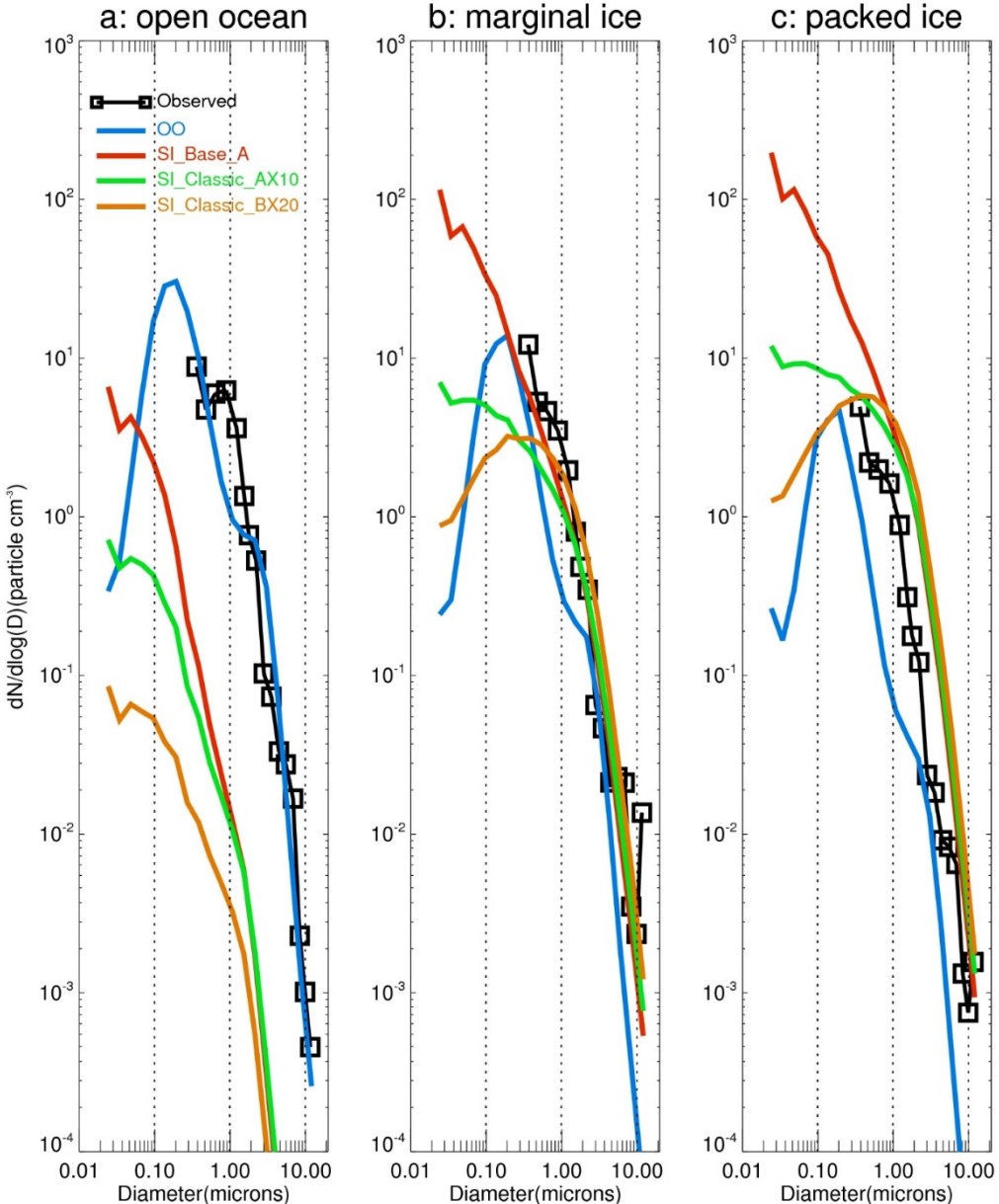

Figure 5: Averaged SSA size distribution from the whole sea ice zone (including both marginal and packed sea ice). Observations are shown in the black line with box symbols. Panel (a) contains model runs with different parameters, including four different sublimation functions (in SI_Base, SI_classic, SI_Area and SI_Mass), two blowing snow size distributions (mode A vs B) and two different ratios of SSA formation per blowing snow particle (in SI_Classic_AX10 and SI_Classic_BX20) (see Table 1 for details). Panel (b) shows model sensitivity to snow salinity. SI_Base_A_SN is same as the control run (SI_Base_A) apart from removing samples with salinity >10 psu. SI_Base_A_SL applies a fixed low salinity of 0.06 psu, and SI_Base_A_SH applies a high value of 0.92 psu. Vertical lines represent diameter of 1 and 10 µm.

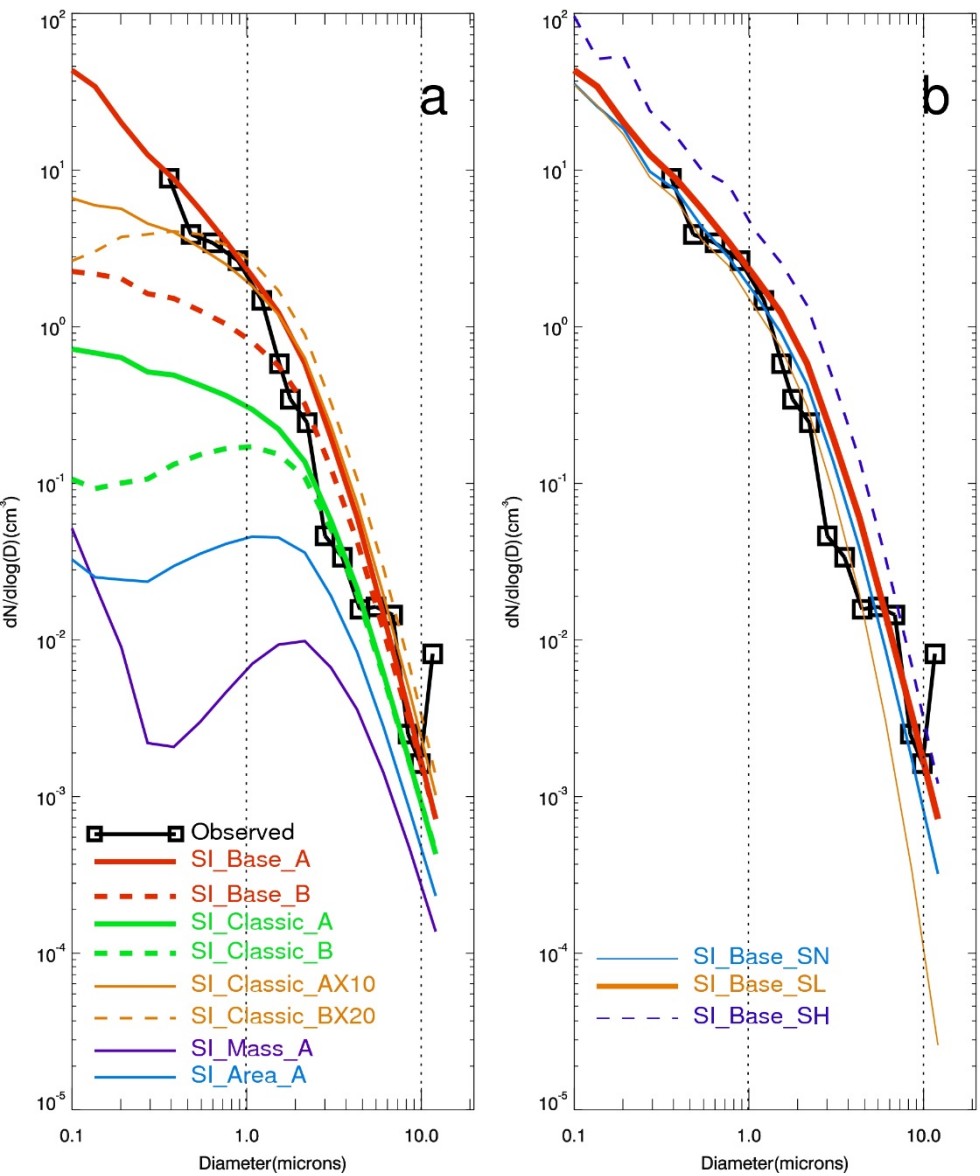

Figure 6: Monthly mean [Na] mass concentration at eight polar sites. Observations are shown in black with diamond symbols, the uncertainty bars representing ±1σ of the inter-annual variability of the observation. Sea spray-derived SSA is shown by the green line (from open ocean control run OO). Sea ice sourced SSA (together with sea spray) from SI_Base_A is shown by the red line with uncertainty bars representing the minimum and maximum of a three year integration (2013-2015). Monthly mean [Na] from SI_Classic_BX20 run shown in orange lines;

5    SI_Base_A_R1 run shown in purple lines; SI_Base_A_R2 run shown in dashed purple lines; and SI_Classic_A run shown in dashed green lines. The mass concentration for model NaCl is at diameter of 0.02-10 µm. The aerosol data are from the following sources: Alert, Barrow and Palmer = AEROCE-SEAREX network (Savoie et al., 2002); Neumayer = Weller et al. (2011); Halley= Rankin et al. (2004); Kohnen = Weller and Wagenbach (2007); Concordia = Legrand at al. (2016); Summit = Mosher et al. (1993) but after Rhodes et al. (2017).

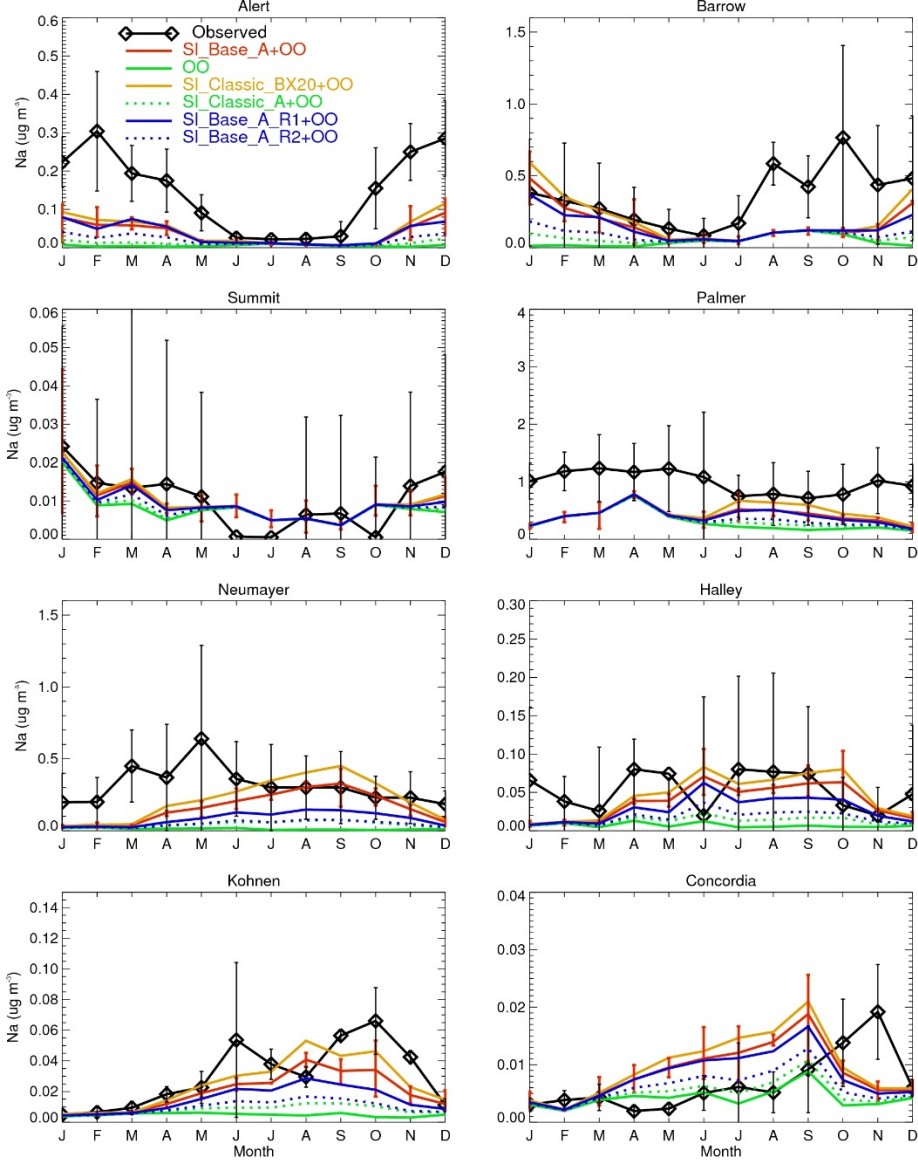

Figure 7: Zonal mean SSA total number concentration (particle cm$^{-3}$) from the sea spray open ocean in (a) months June-July-August (JJA) and (b) months December-January-February (DJF) from OO run. Sea ice sourced SSA from SI_Base_A run shown in c and d, with SI_Classic_AX10 run result in e and f, and SI_Classic_BX20 run result in g and h. The plots are based on one year (2013) integration. The contour interval is 10 particle cm$^{-3}$ when number density is larger than 10 particle cm$^{-3}$.

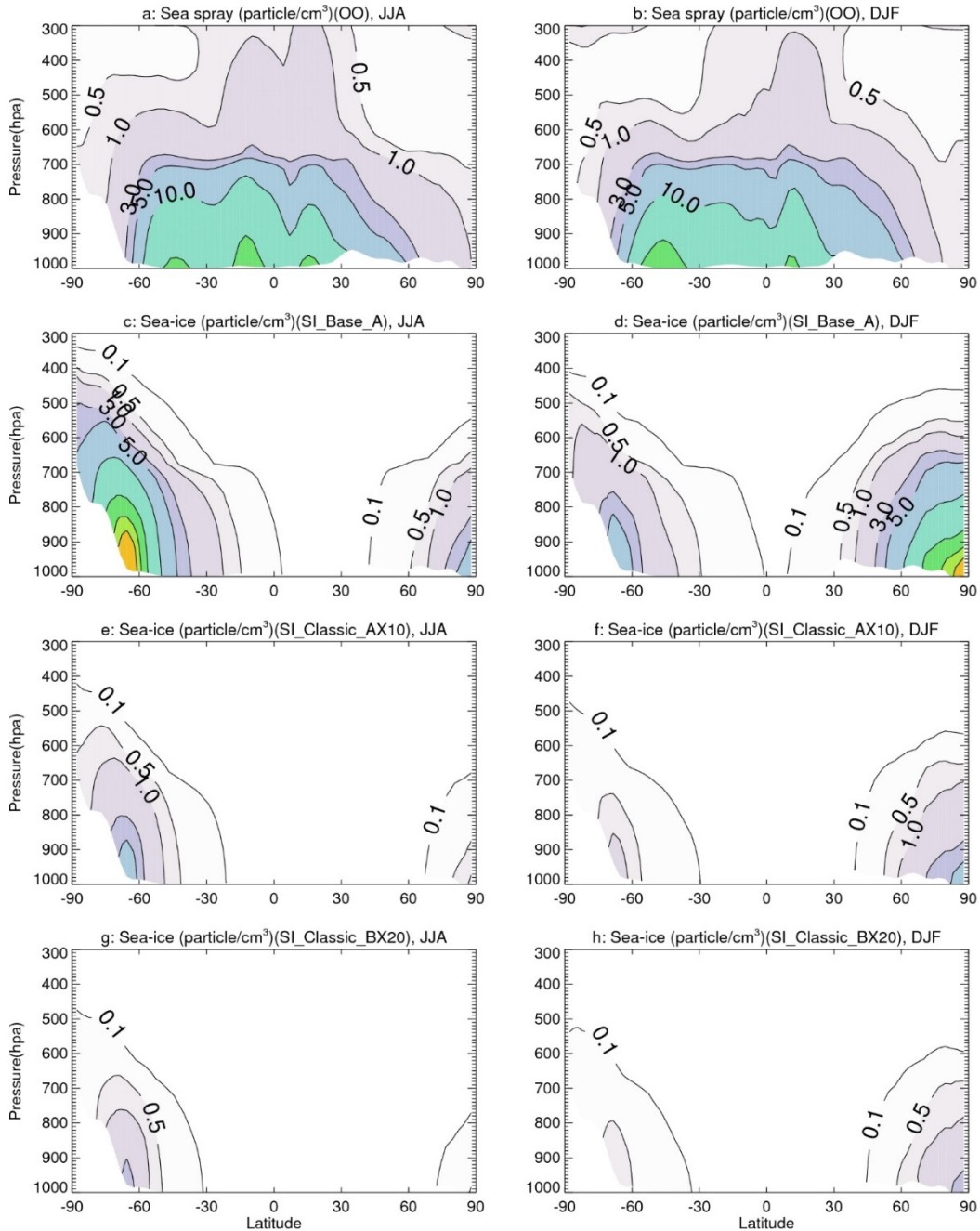

Figure 8: (a) Allocated sublimation fluxes across different snow size bins (with bin interval of 1 μm) in each experiment. Note, the bulk sublimation flux used for allocation is calculated under conditions of wind speed=12 m s$^{-1}$, temperature=-10°C, *RH*(w.r.t. ice)=80%, and snow age=0 day. (b) Converted blowing snow particle production flux. (c) Corresponding SSA number production flux. Note, the conversion is under a fixed snow salinity of 0.06 psu and assuming one SSA from one saline blown-snow particle. Two open ocean sea spray fluxes under the same wind speed of 12 m s$^{-1}$ (SST=5°C for OO_Jaeglé) are shown for comparison. (d) Same as (c) apart from for mass flux. (e) Accumulated mass flux percentages.

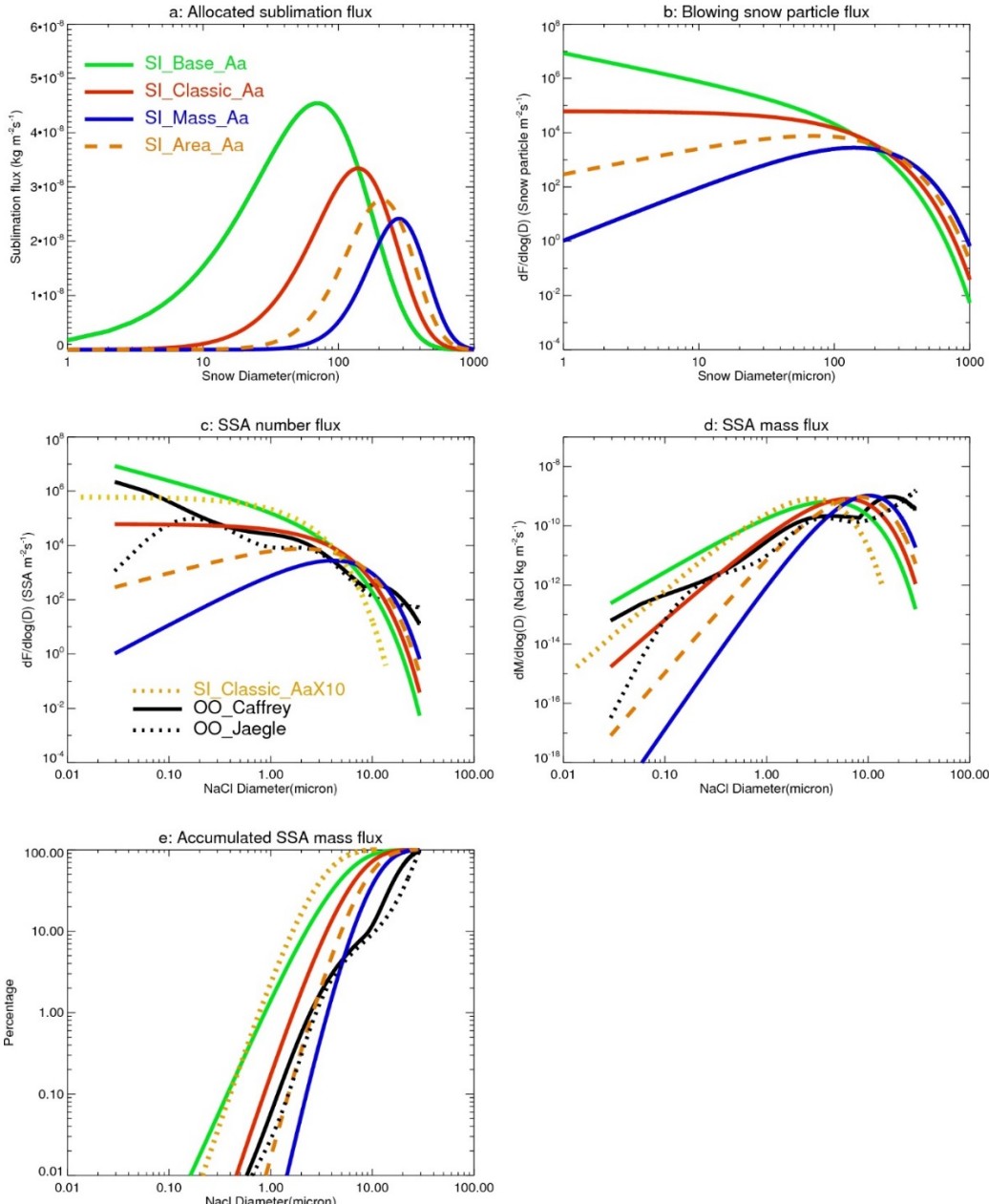