# Peer review of "Sea salt aerosol production via sublimating wind-blown saline snow particles over sea-ice: parameterizations and relevant micro-physical mechanisms"

_Atmospheric Chemistry and Physics, 2018_

## Referee Comment (RC1) · Anonymous Referee #1 · 2 Jan 2019

The comment was uploaded in the form of a supplement:
https://www.atmos-chem-phys-discuss.net/acp-2018-1080/acp-2018-1080-RC1-supplement.pdf

---

## Referee Comment (RC2) · Anonymous Referee #2 · 12 Feb 2019

Comment on "Sea salt aerosol production via sublimating wind-blown saline snow particles over sea-ice: parameterizations and relevant micro-physical mechanisms" by Yang et al.

General comments: This manuscript describes using cruise measurements of snow particles and aerosols particles at Weddell Sea to constraint the parameterizations of sea salt aerosol production from blowing snow events in the p-TOMCAT model. This work conducts several sensitivity model simulations, and suggest that two different micro-physics mechanisms can potentially explain the observations. However, given

the observational data is not available for aerosols <0.4 $\mu$m, the constraints for submicron aerosol is relatively weak, and thus it is hard to determine which of the two micro-physics mechanism is dominant. Overall, this work provides novel insights for the sea salt aerosols production over the polar regions and highlight the necessity of future observations. Therefore, I support the publication of this work in ACP if the following specific comments can be addressed.

Specific comments: P1 Line 23: Does "in size range of 0.4-10 $\mu$m" refer to radius or diameter?

P2 Line 10-13: What is the size converting ratio from blowing snow particles to SSA? Are all/most SSA with size of 0.375-10 $\mu$m being generated by from snow particles in 46-500 $\mu$m?

P7 Line 26: Adding a plot showing the probability distribution of surface snow salinity used in the model can be very helpful. Also, does the probability distribution of surface snow salinity changes in times during the cruise period? For example, is the surface snow more saline in the earlier winter time?

P8 L1: Does the cruise travel mostly in the first-year sea ice area or multi-year sea ice area? Does the measured SSA mostly come from first-year or multi-year sea ice?

P8 L15: If the snow age = 0, does it mean that they are fresh fallen snow? If this is the case, can they get saline in such short time period? Will the surface snow salinity can be substantially lower for those snow age = 0, compared to 1-3 days? Please justify for this assumption of snow age=0.

P9 L17: Please justify why N=10 or 20 are chosen here.

P10 L15: I don't quite understand this sentence. Open ocean contributes to ∼20% and sea ice contributes to 40-50%. Are the rest of observations (20-30%) contribute by non-sea-salt source? Please provide a little bit more details.

P12 L2: As the cruise traveled through marginal sea ice and packed sea ice region, I

am curious if there are any signals in the aerosol samples indicating the salinity difference between these two regions?

P12 L15: As mentioned in previous section (P11 L13), the model runs with reduced RH (SI_Base_A_R1 and SI_Base_A_R1) outperformed the run SI_Base_A, with SI_Base_A overestimating the aerosol number observation. However, from Figure 5, the run SI_Base_A seems to underestimate Na at some of the sites (Alert, Barrow, Palmer). How are the run with reduced RH perform comparing to these global sites?

Also, it seems that the RH is too high in the model, causing the overestimates of blowing snow production. Meanwhile, the model does not consider the drifting snow at low wind speeds, causing underestimates of the model. These two effect seem compensate each other in the model. Please elaborate in a little bit more details in how the model can be better constrained. For example, what types of observations are required or suggested to distinguish the effect of these two factors?

In addition, the comparison between the three model runs here (SI_Base_A, SI_Classic_AX10 and SI_Classic_BX20) are using different evaporation rate. Will the SI_Classic_A perform differently compared to SI_Base_A in the comparison in Figure 5?

Figure 1: Please providing lat/lon and/or location map for this plot if possible.

Table 1: SI_Base_A_T1 is named here, but in the manuscript, it is mentioned as SI_Base_A1_T1. Please check the simulation names for T1, T2 and T3 as well. Also, the model result for SI_Base_A1_T3 is not discussed in the manuscript.

---

## Author Comment (AC1) · 3 Apr 2019

We thank the anonymous reviewer for their thoughtful comments and suggestions. We address each one directly below and outline changes that will be made to the revised manuscript.

**Reply to Anonymous Reviewer #1**

Specific comments:

*Abstract, line 28: Please specify "data production rate of 10 SSA formed from one snow particle".*
Answer: we rephrase that sentence to 'If we assume ~10 SSAs can be generated from one snow particle during the evaporation process, then model could reproduce the observations'

*Line: "very similar results" is too vague, please specify: Something like "Although both mechanisms generate very consistent results with respect to observed aerosol number densities."*
Answer: Thank you, it has been changed according to your suggestion.

*Introduction: You can remove line 30: A brief conclusion is presented in section 6.*
Answer: Done

*Section 3.1: line 23: "from the " appears in bold in my version.*
Answer: corrected

*You said "The control run for open ocean sea spray is SI_Base_OO, following the scheme by Jaeglé et al. (2011)". But this run is denoted OO_Jaeglé in Table 1, right ?.*
Answer: There are three experiments regarding open ocean sea spray in Table 1: OO, OO_Jaeglé and OO_Caffrey. Both OO and OO_Jaeglé apply Jaeglé et al. (2011) scheme, but they are slightly different. OO result is used in cruise data-model comparisons. OO_Jaeglé is only used for Figure 7, as it is driven by fixed meteorology data, e.g., with a fixed wind speed of 12m/s.

*Section 3.3.4: Be carefully (remove) with abbreviations NH (northern hemisphere) and SH (southern hemisphere) since you already used SH for high salinity.*
Answer: Thank you for pointing out this issue. In the revision, we will use 'southern hemisphere' and 'northern hemisphere' to replace SH and NH respectively.

*Section 3.3.7, line 14: Please correct «single»*
Answer: Done.

*Section 4.2: Here or in figure 5 caption, may be good to indicate references for the observations at the different polar sites.*
Answer: The aerosol data are from the following sources: Alert, Barrow and Palmer = AEROCE-SEAREX network (Savoie et al., 2002); Neumayer = Weller et al. (2011); Halley= Rankin et al. (2004); Kohnen = Weller and Wagenbach (2007); Concordia = Legrand at al. (2016); Summit = Mosher et al. (1993) but after Rhodes et al. (2017). References for the observations used in Figure 5 will be given in the revised version.

*Conclusion:*
*I think the sentence "However, the aerosol concentration (Frey et al., in preparation) gradient observed between near surface (~2m above snow surface) and ~29 m will not allow*

*us to conclude robustly where the SSA is produced. "is an important new information that would appear earlier in the manuscript (the conclusion is not exactly the right place for such a new information).Whereas I fully agree with your conclusion "Thus, this highlights the need for further in-situ observations and laboratory investigation to fill this gap.", but it may be nice to be more precise here. For instance, did the study of the size segregated chemical composition of sea-salt aerosol that can cover the range between 0.03and 20 micron diameters can help ?*

Answer: Good suggestion. In the revised version, we discuss this issue in section 5 (Physical mechanism of SSA production from blowing snow) with a new paragraph shown below.

'Cruise data show that blowing snow particle number density decreases significantly, e.g., by more than an order of magnitude from near surface (~2m above snow surface) to ~29 m. However, aerosol number densities between these two levels do not show such a large gradient as observed for blowing snow. For example, observed data indicate (Figure 5 in the companion paper by Frey et al., acp-2019-259) that during drifting snow episodes aerosol number densities increased significantly especially of sub-micron sized particles at both measurement heights, with a lightly greater increase near the surface (number density up to $10^7\,\mathrm{m^{-3}}$ for diameter $<2\mu m$). During blowing snow number densities showed similar increases as during drifting snow, however at 29 m concentrations were higher and particles were larger (at diameter $>9\,\mu m$) than at 2 m. This observational evidence prevents us from deriving any robust conclusion regarding where SSA is generated: in the near surface layer where *RH* is saturated or at the top of the blowing snow layer where *RH* is under saturated. If SSA is mainly produced near the surface layer, then snow particle evaporation will be controlled by the 'curvature effect' (following the SI_Base corresponded mechanism). However, if SSA is produced in the sub-saturated condition, e.g. at the top layer or above the blowing snow layer, then water evaporation will be controlled by the SI_Classic corresponded mechanism.'

Model experiments with the above two mechanisms implemented (e.g., SI_Base_A and SI_Classic_AX10) can produce roughly the same number density at size range of 0.375-10 μm. However, at SSA size $<0.375\mu m$ diameter, their results are quite different as shown in Figure 3c and Figure 4a. For example, at diameter of 0.1 μm, SI_Base_A has a mean SSA number density almost an order of magnitude larger than that of SI_Classic_AX10 and SI_Classic_BX20. Therefore, a precise observation of SSA at sub-micron size mode can help to diagnose which micro-physical mechanism(s) dominates the SSA production. A systematic measurement of the size segregated chemical composition of SSA over a size range of 0.03 to 20 μm diameter, together with a complete spectrum of blowing snow particle size will help to distinguish which mechanism dominates SSA production from blowing snow (also refer to reply to reviewer #2 comments).'

*In addition to extend the information towards the smallest particles, such chemical information (the sodium to sulfate fractionation for example) would permit to investigate the mixing between particles emitted from open-ocean and from marginal ice.*

Answer: When information of non-sea-salt sulfate contribution is well known, the sodium to sulfate fractionation analysis would help in investigating the mixing between particles emitted from open ocean ad from sea ice. Otherwise the $Na/SO_4$ ratio isn't that helpful. Measuring sulfur isotopes on the sulfate maybe needed in addition.

*Figure 1: Please introduce also the green line (open ocean) in the caption.*

Answer: Done

*Figure 3: The vertical scale (10-4to 103) is the same for the three panels so, removing the numbers in panels b and c, would permit to increase the horizontal scale and to better see the difference in the observations between panels a, b, and c. If not (or in addition), please add a vertical dashed line at one micron on the three panels.*

Answer: Vertical dashed lines for diameter of 0.1, 10 and 10 µm have been added to panels in Figure 3 (and Figure 4).

[Figure]

Updated Figure 3.

---

## Author Comment (AC2) · 3 Apr 2019

We thank the anonymous reviewer for their thoughtful comments and suggestions. We address each one directly below and outline changes that will be made to the revised manuscript.

**Reply to Anonymous Reviewer #2**

Specific comments:
*P1 Line 23: Does "in size range of 0.4-10µm" refer to radius or diameter?*
Answer: Size refers to diameter. Text has been updated.

*P3 Line 10-13: What is the size converting ratio from blowing snow particles to SSA? Are all/most SSA with size of 0.375-10µm being generated by from snow particles in 46-500µm?*
Answer: The conversion factor varies as a function of initial snow particle size, snow salinity and the production ratio number N (following equation 11). The added figure shown below indicates calculated corresponding dry SSA diameter (µm) of initial blowing snow particles in size range 1-1000 µm and at snow salinity range 0.0001 - 100 psu (under N=1). As can be seen that for an initial snow particle =10 µm, the corresponding SSAs under different salinity are most in sub-micron size. For an initial snow particle=100 µm, the dry SSAs formed are most <10 µm, apart from at high salinity of a few tens psu. For snow particles in range of 46-500 µm, at salinity of 0.01-0.1 psu (close to the median salinity, see salinity distribution figure in page 5 below), SSAs formed are most in size 0.5-10 µm. At low salinity=0.001 psu, the corresponding SSAs are mainly sub-micron sized. At high salinity=1 psu, SSAs are most micron sized. Note, at N=10, the corresponding dry SSA size will be roughly half of the values for N=1. The above text and the Figure will be included in the revision.

[Figure]

Figure: Equivalent dry SSA diameter (µm) as a function of initial snow particle diameter (µm) and snow salinity (psu) for N=1, calculated following equation 11.

*P7 Line 26: Adding a plot showing the probability distribution of surface snow salinity used in the model can be very helpful. Also, does the probability distribution of surface snow salinity changes in times during the cruise period? For example, is the surface snow more saline in the earlier winter time?*

Answer: The observed snow salinity distribution has been included in the companion manuscript (by Frey et al., acp-2019-259), we thus will not show it in this manuscript, but show it below for your information. As can be seen from the figure, snow salinity over young sea ice is indeed higher than that when the vessel is over multi-year sea ice. Snow salinity also change with time during the blowing snow event, for example, during 10-13 July, 2013, bulk snow salinity increased by a factor of two, likely due to the wind erosion effect.

[Figure]

Figure 12 in the companion manuscript (Frey et al., acp-2019-256): Panel (a) shows salinity Sp of snow on first-year sea ice (FYI, yellow symbols) at ice stations S1-6 and multi-year sea ice (MYI, blue symbols) at ice stations S7-9 in the Weddell Sea during austral winter 2013 as a function of snow layer height above the sea ice surface. For comparison Sp of the sea ice surface (triangles) and blowing snow at 1-17 cm above the snowpack (squares) are shown as well. The vertical dashed line indicates Sp (=35.165 psu) of reference sea water (RSW). Panel (b) shows salinity Sp probability distributions for shallow snowpacks (mean depth 21 cm) above first-year sea ice (FYI) at ice stations S1-6 and for deep snowpacks (mean depth 50 cm) above multi-year sea ice (MYI) at ice stations S7-S9 in the Weddell Sea during austral winter 2013. Panel (c) shows respective cumulative probabilities of Sp.

*P8 L1: Does the cruise travel mostly in the first-year sea ice area or multi-year sea ice area? Does the measured SSA mostly come from first-year or multi-year sea ice?*

Answer: The ship was in or near multi-year sea ice from 24 July to 6 August 2013. Thus, SSA are measured over both first-year and multi-year sea ice. Which is included in the manuscript.

*P8 L15: If the snow age = 0, does it mean that they are fresh fallen snow? If this is the case, can they get saline in such short time period? Will the surface snow salinity can be substantially lower for those snow age = 0, compared to 1-3 days? Please justify for this assumption of snow age=0.*

Answer: Snow age was initially introduced to the parameterization to counteract the relatively high snow salinity used (Yang et al., 2008). At present, this parameter amounts to a crude tuning tool with no clear physical meaning. Snow age =0 gives the largest coefficient (=1) to the production flux, therefore, by setting snow age to zero, we effectively remove this

parameter altogether. 'Snow age' should not be interpreted as the time elapsed after the snowfall.

Actually, the 'snow' here refers to all ice crystals on surface of snow pack that can be mobile or wind-lifted to airborne. These include fresh fallen snow, diamond-dust, wind-cropped frosts or even 'aged' snow that has been re-mobilized by wind-erosion. The mixing of fresh snow and 'old' saline snow changes the salinity distribution, a process has not been considered by the model so far. Due to lack of data, we do not know how fast fresh fallen snow acquires salts. This process may be fast and efficient during a windy condition through physical contact to the salt-rich crystals. With further data, we may have a better representation of this process.

The above text has been included in the revision.

*P9 L17: Please justify why N=10 or 20 are chosen here.*
Answer: The selection of N=10 or 20 was arbitrary and simply based on model experiment trial which gives a good agreement to the observations.

*P10 L15: I don't quite understand this sentence. Open ocean contributes to ~20% and sea ice contributes to 40-50%. Are the rest of observations (20-30%) contribute by non-sea-salt source? Please provide a little bit more details.*
Answer: What the sentence meant is that, in marginal ice, model (a combination of sea spray and sea ice sourced SSA) underestimates the observation by 30-40%. To avoid confusion, we re-phrase the sentence and merge it with the first sentence in the same paragraph: 'In marginal ice (Figure 3b), our simulation suggests that both sea-ice and open ocean sourced SSA are making a contribution to the observations, but they underestimate the observations by 30-40%. As shown in Table 2 that sea spray accounts for ~20% of the observations and sea-ice sourced SSA accounts for 40-50%.'

*P12 L2: As the cruise traveled through marginal sea ice and packed sea ice region, I am curious if there are any signals in the aerosol samples indicating the salinity difference between these two regions?*
Answer: We do not have aerosol salinity data, but aerosol [$Na^+$] concentrations do not show significant different between marginal ice and packed sea ice region. Blowing snow data (in the companion paper) show that 'As expected the mean salinity of the shallow FYI snowpack (=1.4psu) was larger than that of the deep MYI snowpack (0.82 psu), and corresponding salinity probability distributions for snow on FYI are shifted to higher salinities when compared to above MYI.'

*P12 L15: As mentioned in previous section (P11 L13), the model runs with reduced RH (SI_Base_A_R1 and SI_Base_A_R1) outperformed the run SI_Base_A, with SI_Base_A overestimating the aerosol number observation. However, from Figure 5, run SI_Base_A seems to underestimate Na at some of the sites (Alert, Barrow, Palmer). How are the run with reduced RH perform comparing to these global sites? Also, it seems that the RH is too high (low?) in the model, causing the overestimates of blowing snow production. Meanwhile, the model does not consider the drifting snow at low wind speeds, causing underestimates of the model. These two effect seem compensate each other in the model. Please elaborate in a little bit more details in how the model can be better constrained. For example, what types of observations are required or suggested to distinguish the effect of these two factors? In*

*addition, the comparison between the three model runs here (SI_Base_A, SI_Classic_AX10 and SI_Classic_BX20) are using different evaporation rate. Will the SI_Classic_A perform differently compared to SI_Base_A in the comparison in Figure5?*

Answer: In the revised Figure 5 (shown below) we add model results from SI_Base_A_R1 (purple line), SI_Base_A_R2 (dashed purple line) and SI_Classic_A (dashed green line). It can be found from the Figure that SI_Classic_A gives the least SSA production and could not explain the winter SSA observations. This is consistent with the result shown in Figure 4 and Figure 7. The SI_Clasic_A allocates more sublimated water to large blowing snow particles than the SI_Base_A, as a consequence, there are more large SSA (in micron size) and less sub-micron sized SSA being generated. The experiment with a fixed *RH*=90% (with respect to ice) in SI_Base_A_R1 gives a reduced SSA mass concentration comparing to the SI_Base_A (red line), but still shows clear winter SSA peaks in most polar sites. However, the SSA production in SI_Base_A_R2 (at *RH*=95%) is much supressed and will not give a good representation of the winter peaks. Given that mass concentration is dominated by large particles and number density is mainly by small particles, the overestimated SSA number density and the underestimated mass concentration at sites (Alert, Barrow and Neumayer) in SI_Base_A run indicates the current model setups and parameterizations applied need further constraints and evaluation against data. Model runs with higher *RH* values (in SI_Base_A_R1 and SI_Base_A_R2) reduce both SSA number density and mass concentration, thus model moisture will not reconcile the discrepancy between the model and the observations in number density and mass concentration. Inclusion of the missed drifting snow as a source of SSA will add extra SSA to polar sites and affect SSA mass concentration and number density in the same way.

As discussed in section 3 and 4, apart from the evaporation rate, there are other parameters that also affect SSA size spectrum. An outstanding one is blowing snow size distribution: the shape parameter $\alpha$ and the mean diameter (they determine scale parameter beta $\beta$). As shown in Figure 4a, for example, the run with a larger alpha $\alpha$=3 (in SI_Base_B) gives much reduced SSA number densities in submicron size mode comparing to the result in SI_Base_A (where a smaller $\alpha$=2 is applied with same mean diameter). In addition, cruise data show that blowing snow particle size distribution varies as a function of height (above the surface) and wind speed (see companion manuscript by Frey et al.). Therefore it is important to apply more realistic blowing snow distribution to constrain this key parameter. We plan to investigate this issue further by applying a time-series of observed blowing snow size distribution along the cruise track to further constrain this parameter to narrow down the uncertainty. Other parameters such as snow salinity distribution and number N of SSA formed per snow particle may also affect the spectrum. Due to lack of data we could not check them in detail.

[Figure]

Updated Figure 5.

*Figure 1: Please providing lat/lon and/or location map for this plot if possible.*

Answer: The cruise track location map has been given in the companion paper. Here for your information we show it below.

[Figure]

Figure 1 of the companion manuscript by Frey et al.: Cruise tracks of RV Polarstern in the Weddell Sea for the winter expedition ANT-XXIX/6 from 8 June to 12 August 2013 (red line) and the spring expedition ANT-XXIX/7 from 14 August to 16 October 2013 (yellow line). Symbols indicate the location of ice stations S1–9 (Table 1). Crosses show ship positions when entering the sea ice on (A), reaching the marginal sea ice zone MIZ (B) and returning to the open ocean (C). Sea ice concentrations on 15 July 2013 are shown as shaded area, and sea ice extent on 15 June, 15 August and 15 September 2013, respectively, as grey solid lines, all based on Nimbus-7 satellite microwave radiometer measurements (Comiso, 2018).

*Table 1: SI_Base_A_T1 is named here, but in the manuscript, it is mentioned asSI_Base_A1_T1. Please check the simulation names for T1, T2 and T3 as well. Also,the model result for SI_Base_A1_T3 is not discussed in the manuscript.*

Answer: Thank you for pointing out the mistake made. Now they are SI_Base_A_T1, SI_Base_A_T2 and SI_Base_A_T3 in the revised manuscript. We also corrected a misused experiment name 'SI_Base_T3' (Page 11 line 3), it is now 'SI_Base_A_T3'. Thus the missed 'SI_A_T3' is discussed in the text.

---

## Author Response (AR1)

**Sea salt aerosol production via sublimating wind-blown saline snow particles over sea-ice: parameterizations and relevant micro-physical mechanisms**

Xin Yang1, Markus M. Frey1, Rachael H. Rhodes2\*, Sarah J. Norris3, Ian M. Brooks3, Philip S. Anderson4, Kouichi Nishimura5, Anna E. Jones1, Eric W. Wolff2

2 Department of Earth Sciences, University of Cambridge, Cambridge, UK

4 Scottish Association for Marine Science, Oban, Argyll, Scotland, UK
 5 Graduate School of Environmental Studies, Nagoya University, Nagoya, Japan
 \*now at Department of Geography and Environmental Sciences, Northumbria University, Newcastle upon Tyne, UK

Correspondence to: Xin Yang (xinyang55@bas.ac.uk)

5

Abstract. Blowing snowover sea-ice has been proposed as a significant source of sea salt aerosol (SSA) (Yang et al., 2008).

- 15 In this study, based onusing data (e.g.-snow salinitydata and blowing snow and aerosol particle measurements) collected in the Weddell Sea sea-ice zone (SIZ) during a winter cruise, we perform a comprehensive model-data comparison with the aim of validating the proposed parameterizations\_-anAdditionally, we investigateinge possible physical mechanisms involved in SSA production from blowing snow. A global chemicalstry transport model, p-TOMCAT, is used to examine the model sensitivity to key parameters involved, namely blowing snow size distribution, snow salinity, evaporation\_sublimation function,
- 20 snow age, surface wind speed, relative humidity, air temperature and ratio of SSA formed per snow particle. As proposed in Yang et al.'s parameterizations, SSA mass flux is proportional to bulk sublimation flux of blowing snow and snow salinity. To convert bulk sublimation flux to SSA size distribution, requires (1) evaporation sublimation for snow particles, (2) blowing snow size distribution, (32) snow salinity, and (4) ratio of SSA formed per snow particle.
- The best-optimum\_model-cruise aerosol data agreement (in diameter\_size-range of 0.4-102 μm) indicates two possible micro-25 physical processes that could be associated with SSA production from blowing snow. The first one is\_under\_the assumptionsassumes that one SSA is formed per snow particle after sublimation, and snow particle sublimationvaporation is controlled by the curvature effect or the so-called 'air ventilation' effect. The second mechanism allows multiple SSAs to form per snow particle and assumes snow particle sublimation is controlled by the moisture gradient between the surface of the particle and the ambient air (moisture diffusion effect). At a production ratio of ~10, With this latter mechanism. With this

[revised manuscript text omitted]